# Modelling groundwater recharge, actual evaporation and transpiration in semi-arid sites of the Lake Chad Basin: The role of soil and vegetation on groundwater recharge

Christoph Neukum[1], Angela Gabriela Morales Santos[2], Melanie Ronelngar[3], Aminu Bala[4], Sara Ines Vassolo[1]

[1] Federal Institute for Geosciences and Natural Resources, Department of Groundwater and Soil, Stilleweg 2, 30655 Hannover, Germany

[2] University of Natural Resources and Life Sciences, Vienna, Department of Water, Atmosphere and Environment, Institute for Soil Physics and Rural Water Management, Muthgasse 18, 1190 Vienna, Austria

[3] Federal Institute for Geosciences and Natural Resources at Lake Chad Basin Commission, Rond Point des Armes, Ndjamena, Chad

[4] Lake Chad Basin Commission, Rond Point des Armes, Ndjamena, Chad

*Correspondence to*: Christoph Neukum (christoph.neukum@bgr.de)

**Abstract**

The Lake Chad Basin, located in the center of North Africa, is characterized by strong climate seasonality with a pronounced short annual precipitation period and high potential evapotranspiration. Groundwater is an essential source for drinking water supply as well as for agriculture and groundwater related ecosystems. Thus, assessment of groundwater recharge is very important although difficult, because of the strong effects of evaporation and transpiration as well as limited available data.

A simple, generalized approach, which requires only limited field data, freely available remote sensing data as well as well-established concepts and models, is tested for assessing groundwater recharge in the southern part of the basin. This work uses the FAO-dual $K_c$ concept to estimate E and T coefficients at six locations that differ in soil texture, climate, and vegetation conditions. Measured values of soil water content and chloride concentrations along vertical soil profiles together with different scenarios for E and T partitioning and a Bayesian calibration approach are used to numerically simulate water flow and chloride transport using Hydrus-1D. Average groundwater recharge rates and the associated model uncertainty at the six locations are assessed for the 2003-2016 time-period.

Annual groundwater recharge varies between 6 and 93 mm and depends strongly on soil texture and related water retention and on vegetation. Interannual variability of groundwater recharge is generally greater than the uncertainty of the simulated groundwater recharge.

## 1 Introduction

Recharge occurs even in the most arid regions, mainly due to concentration of surface flow and ponding with lateral and vertical infiltration (Lloyd, 1986). Direct recharge by precipitation is possible in semi-arid regions, but intermittently, owing to the fluctuations in the periodicity and volume of precipitation that is inherent to such regions (Lloyd, 2009). Scanlon et al.

(2006) synthesized recharge estimates for semiarid and arid regions worldwide. They found that recharge is sensitive to land
use and cover changes, hence management of such changes are necessary to control recharge. Moreover, they stated that
average recharge rates in semi-arid and arid regions range from 0.2 to 35 mm yr$^{-1}$, representing 0.1 to 5% of long-term average
annual precipitation. Recently, Cuthbert et al. (2019) investigated the relationship between precipitation and recharge in sub-
Saharan Africa using multidecadal hydrographs. They found that focused recharge predominates in arid areas and is mainly
controlled by intense rainfall and flooding events. Intense precipitation, even during years of low annual precipitation, results
in some of the most significant years of recharge in dry subtropical locations.
The arid to semi-arid Lake Chad Basin (LCB) is one of the largest endorheic basins of the world with an area of approximately
2.5 million km². It covers parts of Algeria, Cameroon, Central African Republic, Chad, Libya, Niger, Nigeria, and Sudan.
According to the Lake Chad Basin Commission (LCBC, 2012), 45 million inhabitants are settled in the basin. The study areas
of Salamat and Waza Logone are located in the southern part of the LCB along the Chari-Logone, the major tributary river
system to Lake Chad (Figure 1), which accounts for around 80-90% of the inflow to the Lake Chad (Bouchez et al., 2016).
Groundwater is an important source for drinking water supply as well as for agriculture and groundwater related ecosystems
in the LCB. Lake Chad, associated rivers, and the floodplains of the major rivers are characterized by strong seasonality, due
to a pronounced short annual precipitation period and high potential evapotranspiration. Groundwater recharge, evaporation,
transpiration, and the entire hydrological budget depend strongly on seasonality. However, the impact of transpiration as a
potentially significant process of the hydrological budget (Jasechko et al., 2013) has not yet been intensively explored in the
region (Bouchez et al., 2016).
Many studies have been published concerning the hydrological behaviour and budget of Lake Chad, due to its substantial and
frequent open water surface changes and related consequences to the population and the environment (e.g. Bouchez et al.,
2016; Lemoalle et al., 2014; Lemoalle et al., 2012; Olivry et al., 1996; Vuillaume, 1981). Another important topic associated
to Lake Chad is groundwater recharge by infiltration of lake water into the Quaternary aquifer, which was estimated by isotope
studies (Fontes et al., 1969; Fontes et al., 1970; Zairi, 2008), water and salt budgets (Bouchez et al., 2016; Bader et al., 2011;
Carmouze, 1972; Roche, 1980), and hydrogeological models (Isihoro et al., 1996; Leblanc, 2002).
Very significant for an arid to semi-arid region is the determination of diffuse groundwater recharge, evaporation of surface
and soil water as well as transpiration from plants. In the LCB, recharge has been assessed using different methods. The
Chloride Mass Balance (CMB) approach is a widely used technique. Edmunds and Gaye (1994) used interstitial water chloride
profiles from the unsaturated zone, in combination with measurements of chemical parameters from dug well samples, to
calculate groundwater recharge in the Sahel. They estimated a recharge rate of 13 mm year$^{-1}$ for a mean annual rainfall from
1970-1990 at 280 mm in their study area and concluded that it is an inexpensive technique, which can be applied in many arid
and semi-arid areas. Applying the same method, Edmunds et al. (2002) estimated direct recharge rates from precipitation in
the Manga Grasslands in NE Nigeria (western LCB) at rates between 16 mm year$^{-1}$ and 30 mm yr$^{-1}$. Including Cl values from
dug-wells, they appraised the regional direct recharge for north Nigeria at 43 mm year$^{-1}$, which highlights the importance of
infiltration from precipitation to the groundwater table at the regional scale. Tewolde et al. (2019) applied the CMB on soil
profiles of the LCB, which are partly used in this study. They estimated generally lower annual recharge in Salamat (3 to
111 mm year$^{-1}$) compared to Waza Logone (117 to 163 mm year$^{-1}$), whereas very low values were found for Bahr el Ghazal
(0.2 to 0.8 mm year$^{-1}$) and the northern pool of the Lake Chad (0.6 to 0.8 mm year$^{-1}$). They conclude that one major difficulty
of CMB is the choice of a representative chloride concentration, or the concentration that prevails at greater depths, when
evapotranspiration effects are negligible, particularly for soils with a strong vertical variability in chloride concentrations.
Recharge has also been assessed using groundwater modelling in the LCB (Eberschweiler, 1993; Massuel, 2001; Leblanc,
2002; Boronina et al., 2005; Vaquero et al., 2021), where diffused recharge has been obtained in the process of model
calibration. Calculated values differ considerably, depending on the vertical accuracy of the model and the extension of the
modelled area. It is also possible to determine recharge with the help of isotopes. Goni et al. (2021) used environmental isotopes
to conclude that recharge in the southwestern part of the LCB is mainly the result of strong precipitation events in the middle
of the wet season. Using the Cl$^{36}$ to Cl ratio, Bouchez et al. (2019) estimated a recharge rate of $240 \pm 170$ mm year$^{-1}$ for the
humid part of the LCB in the south. Recharge rates reduce to $78 \pm 7$ mm year$^{-1}$ for areas close to surface water and to $16 \pm 27$
mm year$^{-1}$ for regions unconnected to the hydrological network in the Sahelian part.
Concerning evaporation and transpiration, they were assessed for the Lake Chad coupling hydrological, chemical, and isotopic
models (Bouchez et al., 2016). They conclude, that evaporation varies from $2070 \pm 100$ mm year$^{-1}$ in the southern to 2270
$\pm 100$ mm year$^{-1}$ in the northern pool, whereas transpiration is insignificant with an average of 300 mm year$^{-1}$ in the lake that
increases slightly to 500 mm year$^{-1}$ in the archipielagos, where vegetation is abundant. Furthermore, they state that their work
estimates transpiration of the Lake Chad for the first time. However, studies on evaporation and transpiration in the vadose
zone are largely missing in the LCB.
For vadose zone studies, partitioning evapotranspiration (ET) into its respective soil evaporation (E) and plant transpiration
(T) components is crucial for process-based understanding of fluxes (Anderson et al., 2017). There are a number of
measurement and modelling approaches that can be used to estimate E and T separately, including micro-lysimeters, soil heat
pulse probes, Bowen ratios, and Eddy covariance to determine E, and sap flow, chambers, and biomass-transpiration
relationships to measure T (Kool et al., 2014). Evapotranspiration partitioning can also be estimated directly by using stable
isotopes to assess the ratio between E and T (Wu et al. 2016). Stable isotopes were also used in combination with Eddy
covariance on semi-arid environments (Aouade et al., 2016).
The Food and Agricultural Organization of the United Nations (FAO) published a model (Allen et al., 1998) that uses an
empirically defined crop coefficient ($K_c$) in combination with a grass-reference potential ET ($ET_0$) to calculate crop potential
evapotranspiration ($ET_c$). There are two approaches for this method: single coefficient and dual crop coefficient. The FAO-
dual $K_c$ model is a validated method for ET partitioning and the most commonly applied (Kool et al., 2014). It has been widely
used with good results for numerous crops under different conditions: e.g. wheat and maize in semi-arid regions (Shahrokhnia
and Sepaskhah, 2013), wheat in humid climates (Vieira et al., 2016), cherry trees in temperate continental monsoon climates
(Tong et al., 2016), and irrigated eucalyptus (Alves et al., 2013) and canola in terrestrial climates (Majnooni-Heris et al., 2012).
Quantification of water fluxes in the vadose zone and linking atmospheric water and solute input at the upper boundary of the
soil with water and solute fluxes at different soil depths is frequently implemented using different type of models. Numerical
models need information on vadose zone properties for accurate parametrization to link fluxes with state variables such as
unsaturated hydraulic conductivity and the water retention curve. Estimation of effective soil hydraulic parameters, which are
valid at the modelling scale, might be laborious. Furthermore, parameter estimation might vary significantly depending on the
measurement method (Mertens et al., 2005), when water and solute fluxes dynamics are considered. Hydraulic and transport
parameters obtained from inverse modelling can be ambiguous, if multiple parameters are simultaneously considered and
boundary conditions are not well known. Combining different state variables of water flow and solute transport in one objective
function was found to be a useful strategy for appropriate parametrization (Groh et al., 2018; Sprenger et al., 2015) and for the
transient simulation of water and solute fluxes. However, large amount of data are necessary to obtain accurate estimates of
state variables, which are rarely available in remote areas of Africa, and measurement of related variables are associated with
a huge effort in such environments. Pedotransfer functions (PTF) bridge available and needed data. They are frequently used
to quantify soil parameters (van Looy et al., 2017; Vereecken et al., 2016). PTF strive to provide a balance between data
accuracy and availability (Vereecken et al., 2016). Since PTF usually do not consider soil structure, their results are better for
homogeneous soils than for structured ones (Sprenger et al., 2015; Vereecken et al., 2010).
In general, time series of relevant data for estimating groundwater recharge is scarce in the LCB. A simple, generalized
approach, which requires only limited field data, freely available remote sensing data, and well-established concepts and
models, is tested for assessing groundwater recharge in the semi-arid part of the LCB. This work applies the FAO-dual $K_c$
concept to estimate E and T coefficients at six locations, which differ in soil texture, climate, and vegetation conditions.
Measured values of soil water content and chloride concentrations along vertical soil profiles, partly published by Tewolde et
al. (2019), together with different scenarios for E and T partitioning and a Bayesian calibration approach are used to
numerically simulate water flow and chloride transport as well as to produce time series of recharge. Both measured soil-
moisture and chloride concentrations are necessary for model calibration in order to get reliable estimates for water flow and
solute transport. Average potential groundwater recharge and the associated model uncertainty are assessed for the 2003-2016
time-period. This generalized method is applied to selected sites for estimating recharge in areas with low accessibility, but
cannot be extrapolated to the whole LCB.

**2 Data and methods**

**2.1 Study sites**

The LCB is a Mesozoic basin and a major part of its geology comprises sedimentary formations from the Tertiary and
Quaternary periods (LCBC, 1993). The Quaternary sediments form a continuous layer of fluviatile, lacustrine and aeolian
sands. These medium to fine-grained sands act as an unconfined transboundary aquifer, as do all aquifers in the LCB, and are
isolated from underlying aquifers by a thick layer of Pliocene clay (Leblanc et al., 2007; Vassolo, 2009). The Tertiary formation
(Continental Terminal) consists of sandstones and argillaceous sands and is a classic example of a confined aquifer system
that becomes artesian in the surroundings of Lake Chad (Ngatcha et al., 2008). The availability of water from precipitation as
well as the deposition characteristics of the aquifer play an important role in the recharge of the upper unconfined sands
(Vassolo, 2009).
The study sites (Figure 1, Table 1) are located in the Salamat and Waza Logone floodplains in the southern Sahel zone.
Selection of sites was limited mainly by accessibility and project's goals. The sites correspond to those published by Tewolde
et al. (2019) for these areas, except for site ST4. Site ST4 is located far from any floodplain, which is the focus of this research,
and its soil composition and vegetation are very similar to those from site ST3. Thus, including this location would not provide
any additional information.
The types of soils included in the selected sites are sand, loam, clay, and their combinations, which are the most common in
the LCB. However, they surely do not cover all existent soils, due to the extension of the LCB. Sites ST1 and ST2 in Salamat
as well as WL1 and WL3 in Waza Logone are annually flooded over three months, site WL2 located at the edge of the Waza
Logone wetland is flooded only one month per year whereas site ST3, although close to ST1 in Salamat, is never flooded. In
the Salamat region, mainly sorghum is grown with trees such as Acacia albida, A. scorpioides and A. sieberana, present along
the margins of the floodplains (Bernacsek et al., 1992). In the Waza Logone area, vegetation depends on the duration of
submersion, forming grass savannahs that are flooded for longer periods of time (Batello et al., 2004). The selected sites cover
thus, the most common vegetation in the LCB. Acacia and grass are the most widespread natural vegetation, whereas sorghum
is the most commonly planted corn. Cotton, which is also planted, is only locally produced and generally using irrigation.
Mango trees can be found along the Chari and Logone rivers, but are not representative for the whole basin.
**2.2 Climate data**
Monthly precipitation and potential evapotranspiration data from 1970 to 2019 for the study sites were extracted from the
CRUTS 4 database (Harris et al., 2020). The potential evapotranspiration was calculated using the Penman-Monteith method
and is considered herein as the reference evapotranspiration ($ET_0$). Wind speeds at 10 m above ground for Salamat and Waza
Logone were obtained from Didane et al. (2017). To adjust these values for 2 m above ground, a correction factor of 0.7479
was applied, based on a logarithmic wind speed profile (Allen et al., 1998).
Average annual precipitation in Salamat and Waza Logone are 807 mm and 709 mm, respectively. The rainy season is typically
from May to September with maximum precipitation in July and August. Average annual values of $ET_0$ are 1718 mm in
Salamat and 2011 mm in Waza Logone, exceeding annual precipitation by more than a factor of 2. However, in the second
half of the rainy season, the monthly water balance is positive. The average water balance for July until September between
2003 and 2016 was $131 \pm 101$ mm month$^{-1}$ and $90 \pm 63$ mm month$^{-1}$ for Salamat and Waza Logone, respectively (Figure 2).
Chloride concentration was analyzed in the BGR laboratory in Hanover, Germany using a Thermo Fischer (Dionex) type ICS-
5000 ion chromatograph with a detection limit of 0.003 mg l$^{-1}$. Concentration in ponding water was measured in four samples
in Salamat, which varied between 2.5 mg l$^{-1}$ and 25 mg l$^{-1}$.
Precipitation was sampled using a Hellmann rainwater collector in N'Djamena. This device was designed to minimize to a
minimum evaporation by using a narrow soft polypropylene plastic tube of 4 mm inner diameter to connect the funnel on top
of the device with the bottom of the 3 l collection bottle (Gröning et al., 2012). Once precipitation starts, water rises in the
bottle and into the tube decoupling the atmosphere from the bottle headspace to prevent evaporation. To ensure that evaporation
is as low as possible, sampling took place event-wise. Chloride concentration in precipitation was measured in 59 out of 147
samples collected in N'Djamena between 2014 and 2020 for different precipitation events and stages of the rainy season (Table
S1). Not all rain samples could be analyzed for chloride concentration, due to limited sample volume in minor events at the
beginning and end of the rainy season.
Average chloride concentration in May was $2.5 \pm 2.3$ mg l$^{-1}$ (3 samples). Precipitation in June to September has relatively low
chloride concentrations, declining from $0.6 \pm 0.3$ mg l$^{-1}$ to $0.26 \pm 0.12$ mg l$^{-1}$ and $0.38 \pm 0.14$ mg l$^{-1}$ at the end of the season.
Strong rain events in July and August have chloride concentrations between 0.2 and 0.3 mg l$^{-1}$. The annual wet chloride
deposition sums to $1.8 \pm 0.5$ kg ha$^{-1}$. The measured values are in the range of published data (Goni et al., 2001; Laouali et al.,
2012; Bouchez et al. 2019; Gebru and Tesfahunegn, 2019). Dry deposition of chloride is estimated between 10 – 30% of wet
deposition (Bouchez et al. 2019).
**2.3 Soil and vegetation data**
At each study site, vertical soil profiles were core-drilled using a hand auger. In Salamat, soil profiles were sampled in 2016
(Tewolde et al., 2019) and 2019. In Waza Logone, soil samples were sampled in 2017 only (Tewolde et al., 2019), due to
security reasons in 2019. Each of the soil profiles was sampled in 10 cm intervals and filled into headspace glass vials and
plastic bags.
Each soil fraction was tested for grain size distribution using standard sieving and sedimentation procedures (Tewolde, 2017).
Classification follows the soil texture triangle by the US Department of Agriculture (Šimůnek et al., 2011).
Chloride concentration was analyzed after aqueous extraction from oven dried (105°C for 24 hours) soil samples following
the standard guideline DIN EN 12457-1 (Tewolde, 2017). Chloride concentrations in groundwater, which are used for
comparison, were measured using a DIONEX ICS-3000 ion chromatograph. Data are presented in Tables S2 and S3
Gravimetric water content is the mass of water contained in a sample as a percentage of the dried soil mass. It was obtained
by weighing the moist sample, oven drying it at 105°C for 24 to 48 hours, and weighing it again (Tables S2 and S3). Bulk
densities were not measured in the field, because of the difficulties handling the samples end sending them to the laboratory.
Instead, volumetric water contents were obtained by multiplying the gravimetric water contents for each soil type and location
by typical bulk densities obtained from the Global Gridded Surfaces of Selected Soil Characteristics database (Global Soil
Data Task Group, 2000), although accuracy of the calculated values reduces with sampling depth (Al-Shammary et al., 2018).
The type of vegetation and the annual cycle of crops, length of the flooding period, and vegetation throughout the dry period
were mapped during field work and documented by surveying resident populations. In addition, MODIS vegetation indices
data (Didan, 2015) were used to justify the documented annual cycle of phenology (Figure 3).

## 3 Modelling methodology

Our approach assumes that groundwater recharge is controlled by precipitation, evaporation, and transpiration (surface runoff can be neglected due to the flat topography). Soil moisture and chloride concentration along the soil profile at a certain time are indicators for evaporation and transpiration processes within the root zone. Chloride concentration in soil depends on its input via precipitation and washing out of dry deposition as well as on the amount of evaporation and transpiration on the soil surface and in the root zone. We assume that amount of recharge corresponds to the volume of water that leaves the model profile through the bottom boundary.

The first estimation of evapotranspiration was carried out using the FAO-dual crop coefficient approach that assesses E and T individually. The uncertainty of E and T partitioning on soil water and chloride concentration in the six soil profiles was assessed by considering scenarios of mean, maximum, and minimum E and T coefficients (see 3.1). Calculated time series of E and T for the site-specific vegetation were used to estimate soil water and chloride concentration profiles at the sampling time in each of the six locations using Hydrus-1D. A Bayesian approach was applied to consider uncertainties in chloride concentrations of precipitation and dry deposition, in partitioning E and T as well as in the parametrisation of the soil hydraulic model (Figure 4).

## 3.1 Partitioning of evaporation and transpiration

Evapotranspiration (ET) is the combination of two main processes driven by atmospheric demand: evaporation from the soil (E) and transpiration through the stomata of plants (T) and is an important component of the water balance, especially in semi-arid areas. The FAO provides a model (Allen et al., 1998) for estimating crop evaporation ($ET_c$) based on an empirically defined crop coefficient ($K_c$) combined with a reference evapotranspiration ($ET_0$). Two approaches are possible, single crop coefficient and dual crop coefficient. The latter was applied in this work.

The dual $K_c$ method (Allen et al., 1998) is the sum of two coefficients, the basal crop coefficient ($K_{cb}$) that describes plant transpiration and the soil water evaporation coefficient ($K_e$) that depicts evaporation from the soil surface. $K_{cb}$ is defined as the ratio of crop evapotranspiration over reference evapotranspiration ($ET_c/ET_0$), when the soil surface is dry and transpiration occurs at a potential rate (i.e. unlimited water availability for transpiration). $K_e$ is highest when the topsoil is wet, but diminishes with drying out of topsoil to become zero, if no water remains near the soil surface for evaporation.

The parameters required for the estimation of monthly $ET_c$ are the monthly reference evapotranspiration ($ET_0$), the monthly basal crop coefficient ($K_{cb}$) and the monthly soil water evaporation coefficient ($K_e$):

$$ET_c = ET_0 * K_c = ET_0 * (K_{cb} + K_e) ,  \qquad (2)$$

Onsite information on vegetation and phenology, such as month of planting, full emergence of crops, and harvesting times, was used to define the monthly variation of vegetation at the study sites. These different vegetation periods were combined with crop-specific $K_{cb}$ values for sorghum and grass provided in Allen et al. (1998) for a sub-humid climate with relative humidity of 45% and an average moderate wind speed of 2 m s$^{-1}$. To comply with the local semi-arid climate conditions in

Salamat and Waza Lagone, the coefficients $K_{cb}$ for mid- and late-time vegetation periods were adjusted as proposed by Allen
et al. (1998). Monthly $K_{cb}$ values for Acacia were estimated based on Do and Rocheteau (2003) and Do et al. (2008). Site-
specific monthly variation of ground cover and flooding periods with ranges of crop coefficient ($K_{cb}$), soil water evaporation
coefficient ($K_e$), and root depth are provided in Table S4.

**3.2 Modelling water flow and solute transport**

**3.2.1 Model concept, setup, and initial conditions**

The chloride profiles measured in soil at a certain time represent water and solute budget input from past precipitation events
and can be estimated by transient water flow and solute transport modelling. The model concept assumes that atmospheric
chloride input is restricted to solute in precipitation and that the chloride concentration profile results from solute enrichment
in the soil, due to evaporation and transpiration. An accurate parametrization of the unsaturated flow and transport model as
well as a robust quantification of groundwater recharge are not possible with the available data and hence cannot be included
within the scope of this study. However, the model results estimate groundwater recharge magnitude and variability based on
information regarding soil texture and vegetation as well as associated uncertainty in results. This proposed approach is
appropriate for locations with limited availability of long-term soil water measurements.
The free software package Hydrus-1D version 4.17.0140 was used to simulate transient water flow and solute transport in the
six variably saturated soil profiles. Hydrus-1D numerically solves the Richards (1931) equation for variably saturated water
flow, advection-dispersion equations for heat, and solute transport (Šimůnek et. al, 2009):
$$\frac{\partial \theta(h)}{\partial t} = \frac{\partial}{\partial z}\left[K(h)\left(\frac{\partial h}{\partial z} + \cos\alpha\right)\right] - S(h) \qquad (3)$$
with:
h          soil water pressure head [L]
θ          volumetric water content [$L^3L^{-3}$]
t          time [T]
z          spatial coordinate [L] (positive upwards)
S          sink term [$L^3L^{-3}L^{-1}$]
α          angle between flow direction and vertical axis
K(h)       unsaturated hydraulic conductivity function [$LT^{-1}$]

The processes simulated at the six study sites were water flow, solute transport, and root water uptake. Hydrus-1D requires
input data at daily time steps, but available precipitation and evaporation data were monthly. Daily values were calculated
dividing monthly data by month-specific days. Thus, all days in a month had the same precipitation rate and the same
evapotranspiration rate. Model execution ended at the soil sampling time (December 2016 and July 2019 for Salamat and June
2017 for Waza Logone). Progressive root growth was considered in all profiles except for ST2, in which the roots of the Acacia
trees were distributed along the whole profile and assumed invariant over the simulation period. Since initial conditions of soil
moisture and resident chloride concentration are unknown, arbitrary values were adopted. To account for different residence
times of water and chloride, due to different degrees of evapotranspiration and unknown initial conditions, each model was
run over a period of time long enough to allow the exchange of at least one water column volume. Thus, total modelling periods
are different depending on the soil type at each site: ST1, ST2 start in 1910, which leads to a maximum residence time (MRT)
of 106 years; ST3 in 2010 (MRT = 6 years), WL1 and WL2 in 1990 (MRT = 26 years), and WL3 in 1970 (MRT = 46 years).
All profiles were discretized into 101 nodes and different horizons according to the soil types interpreted from the individual
grain size distributions.

## 3.2.2 Water flow

For calculation of water retention ($\theta$) and unsaturated hydraulic conductivity functions ($K(h)$), the Mualem-van Genuchten
(MVG) model (van Genuchten, 1980) was applied:
$$\theta(h) = \begin{cases} \theta_r + \dfrac{\theta_s - \theta_r}{[1+|\alpha h|^n]^m} & h < 0 \\[2em] \theta_s & h \geq 0 \end{cases} \tag{4}$$


$$k(h) = k_s S_e^{-1} \left[ 1 - \left( 1 - S_e^{l/m} \right)^m \right] \tag{5}$$
where:
$$m = 1 - \frac{1}{n}; \qquad n > 1$$
$$S_e = \frac{\theta(h) - \theta_r}{\theta_s - \theta_r}$$
with
$\theta$        water content [L$^3$ L$^{-3}$]
h        hydraulic head [L]
$\theta_r$        residual water content
$\theta_s$        saturated water content
$\alpha$        inverse of the air-entry value, empirical [L$^{-1}$]
n        pore-size distribution index, empirical [-]
$l$        pore-connectivity parameter, empirical $\approx$ 0.5 [-]
S$_e$        effective saturation [-]
k$_s$        saturated hydraulic conductivity [LT$^{-1}$]

To reduce computational effort, the initial parametrization of these functions was realized using pedotransfer functions
implemented in Rosetta (Schaap et al., 2001), which is a dynamically-linked library coupled to Hydrus-1D. The input
parameters for each profile were the percentages of sand, silt, clay, and bulk density at several depths. Whenever consecutive
layers of a profile showed almost the same grain size distribution (texture) and soil moisture, the layers were lumped together
and parameter averages were used in the model. The tortuosity parameter l [-] of the MVG was set to 0.5 as proposed by
Mualem (1976).
The upper boundary condition was defined as a variable atmospheric condition, whereas the lower boundary was set to zero-
gradient with free drainage of water for all profiles, except WL3 where confined groundwater conditions prevailed below the
confining clay layer encountered at 3.9 m depth. During drilling, groundwater was hit at 3.9 m depth, but rapidly rose to 2.6 m
below surface. Consequently, a constant head condition was implemented at 2.6 m depth.
**3.2.3 Root water uptake and root growth**
The sink term ($S$) in the Richards' equation, defined by Feddes et al. (1978) as the volume of water removed from a unit
volume of soil per unit time due to plant water uptake, was considered in all soil profiles according to the prevailing vegetation
(Table S4). The Feddes' default parameters for grass were used in the ST3 and Waza Logone profiles. In ST1, where sorghum
was planted, Feddes' parameters for corn were used because sorghum is not available in the list. According to Righes (1980)
sorghum and corn roots extract water from approximately the same soil depths and have similar average root density
distribution.
An average root depth of 1 m was adopted in ST1 for the initial and end seasons, and 2 m for development and mid seasons.
In the case of Acacia in ST2, the adopted parameters correspond to deciduous trees. The root depth of the Acacia tree was
considered as constant over the entire simulation period with maximum root distribution at 0.5 m and decreasing distribution
down to 2 m (Beyer et al., 2016). In ST3, the vegetation was defined as grass, while in WL1, WL2 and WL3 it was defined as
grass with a flooding period of 3 months in WL1 and WL3, but only one month in WL2. Rooting depth values used at these
sites range from 0.1 m to 0.5 m, depending on the growth stage of grass. The median maximum rooting depth value of annual
grass in water-limited ecosystems is 0.37 m with a 95% confidence level in an interval of 0.26 m-0.55 m (Schenk and Jackson,

314     2002).

**3.2.4 Solute transport**
The chloride concentration in soil water was simulated using an equilibrium advection-dispersion model implemented in
Hydrus1D. Hydrodynamic dispersion was implemented considering dispersivity values of 1/10th of the individual layer
thickness, a molecular diffusion coefficient of $1.3 \times 10^{-9}$ m²s$^{-1}$, and a tortuosity factor as defined by Millington and Quirk
(1961). Adopted dispersivity values are within reported ranges of 0.08 m to 0.20 m (Vanderborght and Vereecken, 2007;
Stumpp et al, 2009, 2012).
A time-dependent concentration boundary condition was applied to the upper boundary and a zero-gradient boundary condition
to the lower boundary. The transient liquid phase concentration of infiltrating rainwater follows measured chloride
concentration in precipitation sampled in N'Djamena. The chloride concentration of ponding water correspond to four values
measured in Salamat that range from 2.5 mg l$^{-1}$ to 25 mg l$^{-1}$ with an average of 9 mg l$^{-1}$. Initial chloride concentration in soil
water was set to 0 mg l$^{-1}$. However, each model was run over a period of time long enough to allow the exchange of at least
one water column volume (3.2.1). The model does not consider root solute uptake.

**3.2.5 Crop evapotranspiration scenario definition**

Since crop evapotranspiration was not measured, values were simulated using $K_{cb}$, $K_e$, and root depth instead. Because these
parameters are given in ranges (Table S4), seven scenarios with different combinations of $K_{cb}$, $K_e$, and root depth were
developed to assess ranges of crop evaporation (Table 2). Scenario "Mean" corresponds to the average value of all parameters.
Scenarios "Min" and Max" combine the minimum and maximum values, respectively. Scenario "Mix-1" combines minimum
$K_{cb}$ with average $K_e$ and root depth, scenario "Mix-2" minimum $K_e$ with average $K_{cb}$ and root depth whereas scenario "Mix-
3" combines minimum root depth with average $K_e$ and $K_{cb}$.

**3.2.6 Bayesian model calibration**

Based on the crop evapotranspiration scenarios, the models were calibrated and model uncertainty was estimated using a
Bayesian calibration. Bayesian analysis is a combination of the data likelihood and the prior distribution using the Bayes
theorem (ter Braak and Vrugt, 2008). The sum of likelihood functions for soil moisture and chloride concentration was
implemented to calculate the log-likelihood of a simulation given the observations and standard deviations at each calibration
step. The posteriori parameter distribution was estimated using the Differential Evolution Markov Chain Monte-Carlo (DE-
MCzs) algorithm with three sub-chains (ter Braak and Vrugt, 2008) implemented in the R package BayesianTools (Hartig et
al., 2019). The number of iterations was defined individually according to a Gelman-Rubin reduction factor < 1.2.
In the calibration, scaling factors ranging from 0.75 to 1.25 for the MVG parameters (saturated volumetric water content,
alpha, and n) were adopted individually. However, ranges for the MVG model parameter n were constrained to n > 1.01. Log-
transformed saturated hydraulic conductivity for each layer was considered with ranges from -0.5 to 0.5. The scaling factor
for transpiration was simultaneously used as a divisor for evaporation to remain within the calculated rate of $ET_0$. From all
accepted model runs, 100 were randomly selected at each individual location to evaluate average model results and standard
deviations.

## 4 Results

### 4.1 Grain size distribution

Soil textures were defined based on grain size distributions of the six profiles (Figure 5) according to the US Department of Agriculture soil texture triangle. Most profiles are fine-grained soils (clay, sandy clay) and fine-grained soils with intercalation of thin sand and loam layers. Only soil profile ST3 is dominated by sand and sandy clay loam.

### 4.2 Model parametrization

The calibrated parametrization of the MVG model for each layer of the six sampling locations is plausible (Table 3). The posterior distributions of the Bayesian calibration show the sensitive parameters of the model fit. For ST1, these are n, $\theta_s$, chloride concentration, and the transpiration fraction in evapotranspiration (T), but the $k_s$ is less sensitive (Fig. S1). For ST2, the sensitivities of the model parameters are similar with $k_s$ of the upper layer being the most sensitive and chloride concentration the least sensitive (Fig. S2). The model fits of the data from site ST3 are generally insensitive. Only α, n, and $k_s$ of the upper layer as well as chloride concentration show tighter posteriori distributions (Fig. S3). For site WL1, the model parameters n of layers 1, 2, and 3 as well as the saturated water content of layers 3 and 5, and subordinately of layer 4, are sensitive (Fig. S4). For WL2, the model parameters n of all layers, $k_s$ of layer 3, and $\theta_s$ of layers 2 and 3 are sensitive (Fig. S5). For WL3, $\theta_s$ of layer 2, $k_s$ of layers 1 and 2, and the fraction of transpiration in evapotranspiration are sensitive (Fig. S6).

### 4.3 Soil water content, chloride concentration and groundwater recharge

Measured and simulated water content and chloride concentration profiles for individual scenarios are shown in Figure 6. The average root mean squared error (RMSE) of simulated water content for all individual scenarios ranges from 0.02 to 0.06 cm$^3$ cm$^{-3}$ (Table 4). In general, the models reproduce well the water content and chloride concentrations. However, the dynamics of measured and simulated water contents differ considerably for ST1 and partly for ST2, although maximum values do match. This is due to the long chloride residence time in both locations (109 years) in comparison to the length of data availability (49 years for precipitation and 6 years for chloride concentrations). The models do not match the high chloride concentrations in the uppermost part of soil profiles for ST3, WL1, and WL2. The standard deviations in chloride concentration of the randomly selected model runs are exceptionally high in the lower part of ST2 that corresponds to the poor sensitivity of the chloride concentration at the upper boundary and the comparably wide range of measured chloride concentration in ponding water in the Salamat region (2.5 mg l$^{-1}$ – 25 mg l$^{-1}$).

Measured chloride concentrations in groundwater are much lower compared to those in soil profiles (Tables S2 and S3). This is because groundwater encountered in the study area has been recharged in regions, where chloride input does not play an important role. Large amounts of recharge for the Quaternary aquifer occur mainly in the southern part of the Lake Chad Basin, where long-term annual precipitation reaches values over 1000 mm. Any chloride accumulated in the soil is well diluted and washed away periodically.

However, the large differences in chloride concentrations between soils and groundwater demonstrate the enormous accumulation capacity of soils in the Lake Chad Basin, which act as a buffer over years until precipitation is high enough to dilute the profile. This effect is depicted by the different chloride concentrations measured in profile ST1 between 2016 and 2019 (Table S2) and by the model results for profile ST2 (Figure 7).

The interannual variability of modelled groundwater recharge differs considerably among locations (Figure 7, Table 5). In general, interannual groundwater recharge variability depends on vegetation and soil texture with related water retention capacity. Vegetation with deep roots on soil with comparably high water retention capacity have a stronger interannual variability, e.g. at ST1, ST2 where recharge occurs only in years with high precipitation. Fine textured soils with shallow rooting vegetation have an intermediate variability (WL1, WL2, and WL3), where years without recharge occur only during drought periods. The coarser textured soils with grass cover have low interannual recharge variability (ST3) and recharge occurs each year. Years with high precipitation, e.g. 2006, 2007, and 2008 in Waza Logone as well as 2010 in Salamat, produced strong groundwater recharge.

The highest average annual recharge (93 mm) was calculated for ST3 in Salamat (Table 6), where the water balance during the rainy season (July-September) is higher compared to the Waza Logone region, and where shallow rooting vegetation prevails on comparably coarse soil texture with low water retention capacity and higher hydraulic conductivity. The other locations in Salamat have lower calculated annual recharge, due to deep rooting vegetation and higher soil water retention capacity. The impact of soil texture on annual groundwater recharge becomes apparent by comparing the three locations in Waza Logone with the same vegetation on soils with different water retention capacities and hydraulic conductivities. Groundwater recharge expressed as a fraction of precipitation is between 1% and 4% (Table 5), which is within the range of 0.1 to 5% published by Scanlon et al. (2006). Only at WL2 (8%) and ST3 (12%), where coarse soil textures enhance recharge, a comparably high fraction is estimated.

Simulated chloride concentration and water budget of the soils over the simulated time-period are rather unstable and differ for the six locations. At location ST2 with clay loam soil covered by Acacia and grass, accumulation of chloride takes place over several years, due to the high transpiration related to the effective field capacity. However, in high precipitation years, most of the accumulated chloride is leached to groundwater and soil concentration diminishes, which can be seen from the time-varying differences of the cumulative solute fluxes between the top and bottom boundaries (Figure 8). The difference of cumulative solute flux at the top and bottom boundaries represents the magnitude of chloride accumulation in soil. It should be noted that at this site, the measured chloride concentrations cannot be reconstructed, if only input via precipitation is considered. The measured profile can only be plausibly modelled with an additional input via ponding water. Chloride input at the upper boundary is consequently six-times higher at ST2 compared to the other locations considered in this study.

At location ST3, the chloride accumulation is much lower compared to the other locations. The chloride budget is controlled by the fast groundwater recharge response to precipitation, which flushes chloride annually from the soil towards the groundwater. Most of the chloride that infiltrated with precipitation remains in the vadose zone over several years and is

leached towards groundwater mainly during years with precipitation or water infiltration above threshold values (Figure 8).
Chloride accumulation is highest in profiles with clay soils and high effective field capacity (ST1, WL1, and WL3).
Chemical memory effects are subject to the dynamics of the water and chloride balance. Therefore, steady-state assumptions
are unsuitable. Accurate estimations are only possible with transient assumptions.

**4.4 Evaporation and transpiration**

The amount of transpiration depends on the availability of water in the root zone and the type of vegetation cover. At ST1,
annual transpiration presents two peaks: one related to sorghum and the other to grass (Figure 9). At each location and in every
simulation year, soil water content in the root zone reaches the wilting point defined by the specific parametrization of the root
water uptake model.
The actual evaporation rate depends mainly on the availability of water in the upper soil zone (Table 6), but calculated values
are in accordance with other studies in the area (Bouchez et al., 2019). Clay and clay-loam with relatively high water storativity
have larger amounts of evaporated water compared to sand and loam soils. During dry seasons, the uppermost part of the soils
dries up annually, which significantly restricts evaporation.
Actual evapotranspiration is lower than the reference evapotranspiration most of the year. During and shortly after the rainy
season, when sufficient soil water is available, actual evapotranspiration is comparable to or higher than $ET_0$ depending on the
vegetation.

**5 Discussion**

Soil texture information is helpful to constrain the MVG parameter ranges while searching for realistic parameter sets
(Sprenger et al., 2015). However, poor representation of soil moisture dynamics using MVG parameters derived using Rosetta
are reported (Sprenger et al., 2015) suggesting that soil structure has to be taken into account (Vereecken et al., 2010),
especially for soils where high rock content influences water flow due to inherent heterogeneity (Sprenger et al., 2015). The
soils at the locations considered in this study belong to Quaternary sediments in the Lake Chad basin and heterogeneity due to
rock fragments is largely absent. Furthermore, soil moisture dynamics over the year are much higher in soils of the Waza
Logone floodplain compared to soils from the more humid regions in the south, where annual precipitation, although high,
occurs only over 4-5 months. It is expected that high soil moisture dynamics, rather homogeneous soils, and the monthly
resolution of climate data result in a minor impact of soil structure on MVG parametrization and groundwater recharge as
shown in Section 3.2. Soil moisture dynamics at all locations considered in this study are limited by water availability for
evaporation in the uppermost part of the soil and by water uptake in the root zone, but not by the reference evapotranspiration.
However, because time resolution of precipitation and evapotranspiration data is monthly, the models probably underestimate
soil moisture dynamics.
Calculated chloride concentrations for the soil profiles give indications of appropriate MVG parametrization as well as
evaporation and transpiration partitioning. However, uncertainty of chloride input and its transient variability in particular is
expressed in rather wide and partly bimodal distributions of the scaling factor (sc_Conc) included in the calibration (Figures
S1-S6 in supplement material). On one hand, measured chloride concentrations in precipitation are in agreement with other
studies in central Africa (Goni et al., 2001; Laouali et al., 2012; Gebru and Tesfahunegn, 2019) and its transient behaviour
within the rainy season is considered in the applied model. On the other hand, impact of dry deposition is unknown, because
of data scarcity and potential lateral flow of periodic flooding. Furthermore, due to the monthly resolution of the atmospheric
boundary condition, extreme rain events that cause surface runoff cannot be reflected in the model. The variability of chloride
concentration in some of the soil profiles, which cannot be completely reproduced by the model, indicates either a higher
variability of chloride input and/or a larger variability in soil physics.
Bouchez et al. (2019) identified a chloride deficit between deposition and river export in the Chari-Logone river system of
88% (only 12% of the deposited chloride is exported via river water). They refer to the chemical memory effect, which can
play an important role in arid regions. Our simulations show the importance of the vadose zone for storage of chloride over
longer periods of time, which explains the fate of chloride in the basin and confirms the chemical memory effect. In this
context, it must be noted that the thickness of the vadose zone at the locations considered in this study is between 4 m and
21 m, where important amounts of chloride can be potentially stored leading to a strong delay of the chemical signal from
precipitation to groundwater.
Time-dependent recharge cannot be verified with groundwater hydrographs, because these data are not available in the study
area. However, the calculated mean annual groundwater recharge values are within the ranges of 0.2 to 35 mm yr$^{-1}$ estimated
by Edmunds et al. (2002) using the CMB method in seven chloride profiles in northern Nigeria. The larger values (90 mm yr$^{-1}$
in ST3 and 54 mm yr$^{-1}$ in WL2) are due to local coarse soil and fall within the values estimated by Bouchez et al. (2019),
who, based on $^{36}$Cl and Cl budgets in groundwater, propose recharge values between 16 mm yr$^{-1}$ and 240 mm yr$^{-1}$.

## 464  6 Conclusions

The quantitative estimation of groundwater recharge in the LCB is difficult due to the scarce data availability and the expected
low recharge quantities. Estimation of low recharge amounts in arid and semi-arid areas are usually ambiguous, because the
inherent measurement inaccuracies lead to uncertainties during data processing and modelling. Quantification of water and
solute fluxes in the vadose zone is often implemented using long-term time series of soil moisture, pressure heads, and
concentration data in combination with appropriate models. Monitoring of soil moisture and solute concentration over longer
periods at different depths and sites is difficult in the LCB, due to limited infrastructure and challenging climatic boundary
conditions. The presented approach combines soil moisture and chloride concentration quantified along vertical soil profiles
in different locations within the LCB with numerical models and freely accessible data, while considering data uncertainty.
Calculated chloride concentrations for the soil profiles provide appropriate MVG parametrization as well as evaporation and
transpiration partitioning. Although measured and simulated dynamic behaviour of both water contents and chloride
concentrations differ considerably in profiles ST1 and partly in ST2, their magnitudes largely agree. This is especially
important for chloride concentrations in the middle and deeper parts of the profiles, where seasonal effects are mainly averaged.
Thus, the estimates of soil water balance and especially of groundwater recharge as well as the adopted soil physical parameters
are plausible.
Mean groundwater recharge values estimated in this study are different from those published in Tewolde et al. (2019). This is
due to the more extensive availability of chloride concentration data in precipitation for this study. In addition, Tewolde et al.
(2019) roughly estimated one value of saturated porosity for each profile. This parameter is rather sensitive in the Bayesian
calibration and several values along each of the profiles were considered in this study. In contrast to the assessment of
groundwater recharge with the CMB (Tewolde et al., 2019), the method used here allows not only estimates of mean recharge,
but also its interannual dynamics, variability, and the classification of the uncertainties of the input data and modelling results.
The interannual variability of groundwater recharge is generally higher than the uncertainty of the modelled groundwater
recharge. The soil moisture dynamics at all locations considered in this study are limited by water availability for evaporation
in the uppermost part of the soil and by water uptake in the root zone and not by the reference evapotranspiration.
Simulations show the importance of the vadose zone for storage of chloride over longer time-periods and explain the fate of
chloride in the basin. The thickness of the vadose zone at the locations considered in this study varies between 4 m and 21 m.
Important amounts of chloride can be potentially stored significantly delaying the chemical signal from precipitation to
groundwater.
Upscaling of the results to larger areas must be interpreted with caution since the considered combinations of soils and
vegetation probably do not cover all combinations present in the Salamat and Waza Logone regions.

## Code availability

The software HYDRUS 1D belonging to the U.S. Department of Agriculture was used. The latest version of the program can
be downloaded from https://www.ars.usda.gov/research/software/download/?softwareid=97&modecode=20-36-15-00.

## Author contribution

M.R. conducted fieldwork; A.G.M.S. and C.N. conducted modelling and interpretation; C.N. and S.V. designed the study and
completed the writing. All authors contributed to the discussion of results and commented on the manuscript.

## Competing interests

The authors declare that they have no conflict of interest.

## Acknowledgement

This study was conducted within the framework of the technical cooperation project "Lake Chad Basin - Management of
Groundwater Resources" jointly executed by the Lake Chad Basin Commission (LCBC) and the German Federal Institute for
Geosciences and Natural Resources (BGR). The technical project is funded by the German Federal Ministry for Economic

Cooperation and Development (BMZ). We thank Daniel Tewolde, Paul Königer and Anna Degtjarev for their support in the lab. We are highly indebted to John Molson for the thorough linguistic review of our manuscript.

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

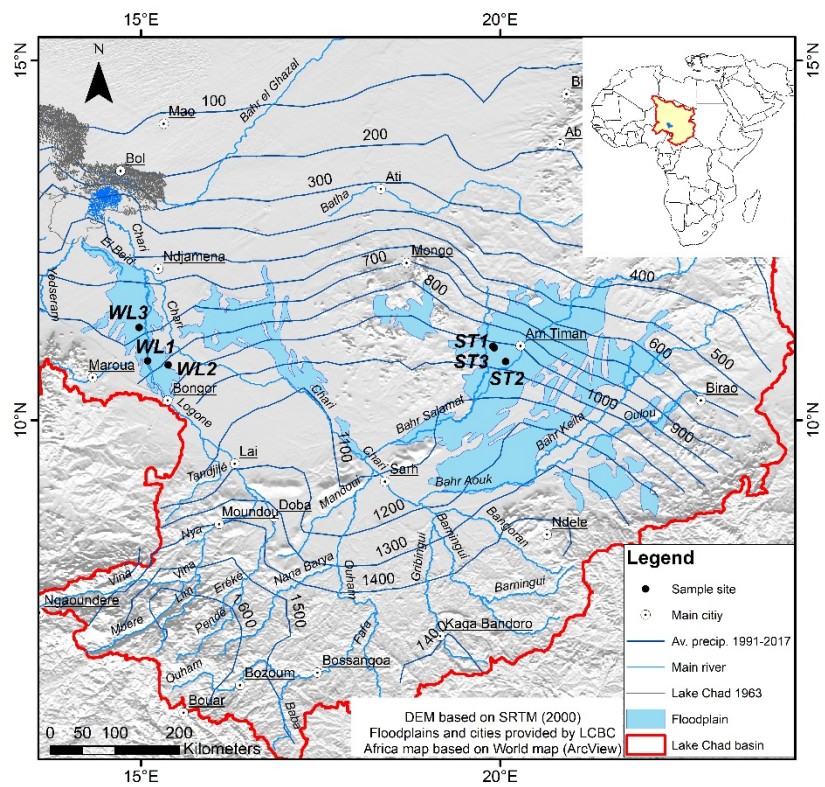


**Figure 1: Location of the six soil sampling sites within the Logone and Salamat river basins in the Lake Chad catchment. The map**
**inset shows the location of the Lake Chad basin in Africa.**

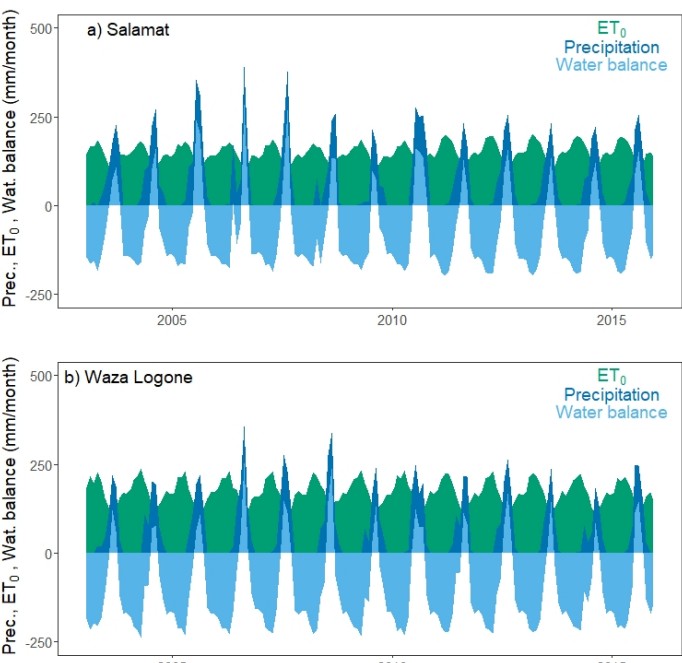


**Figure 2: Monthly precipitation, reference evapotranspiration from the CRUTS 4 database (NCAR, 2017) and derived water balance for Salamat and Waza Logone.**


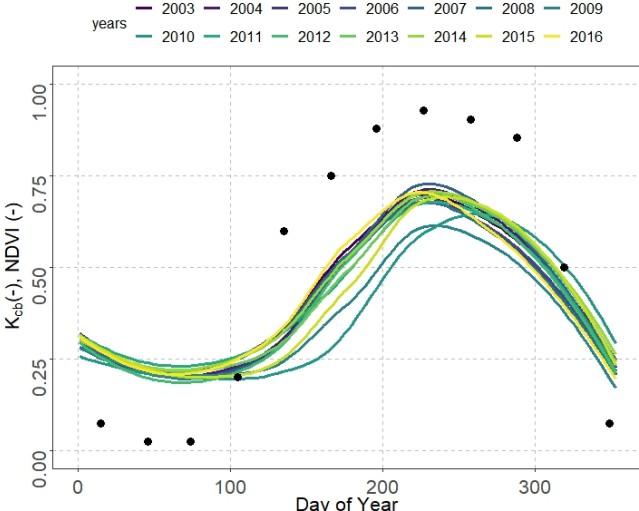


**Figure 3: Average Normalized Difference Vegetation Index (NDVI, MODIS 16 day interval and 250 m spatial resolution) measured between 2003 and 2016 in the Salamat region and estimated monthly basal crop coefficient ($K_{cb}$, black points) for location S3.**



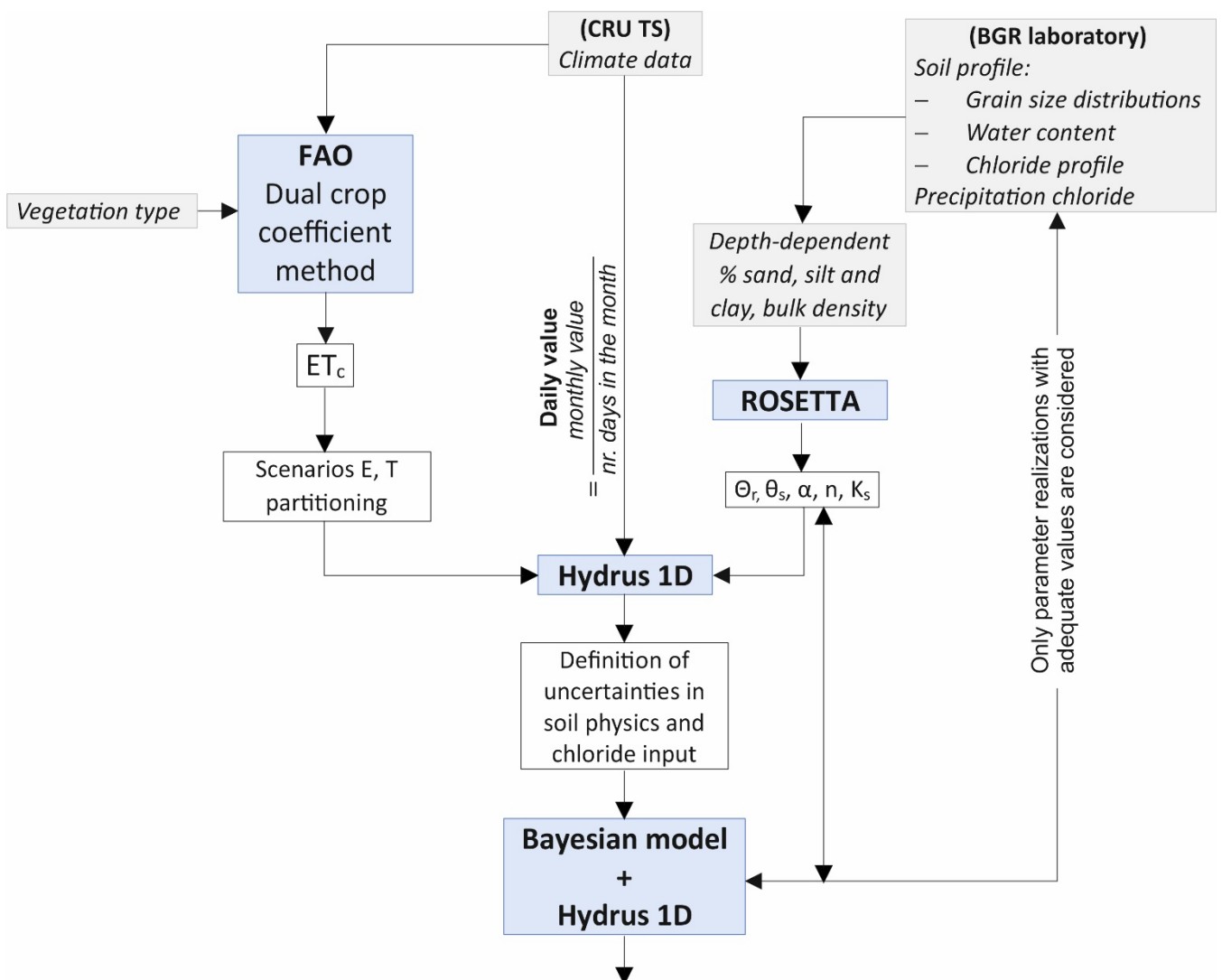


**Figure 4: Workflow of the modelling activities**

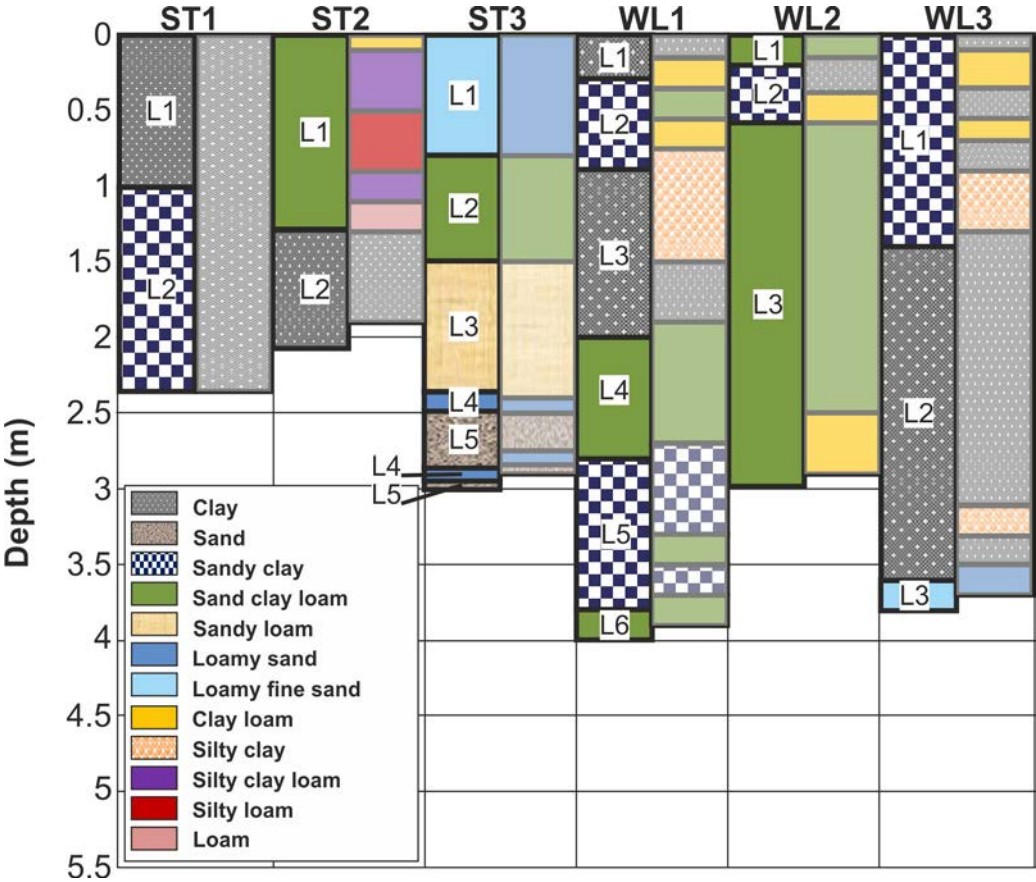

**Figure 5: Soil textures used in the model (left column) defined according to the grain size distribution analysis (right column) for**
**each of the six soil profiles.**

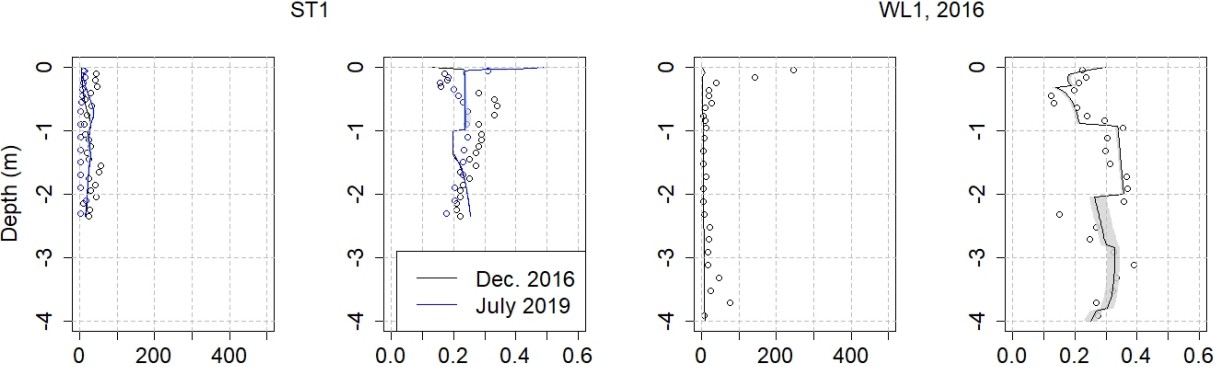


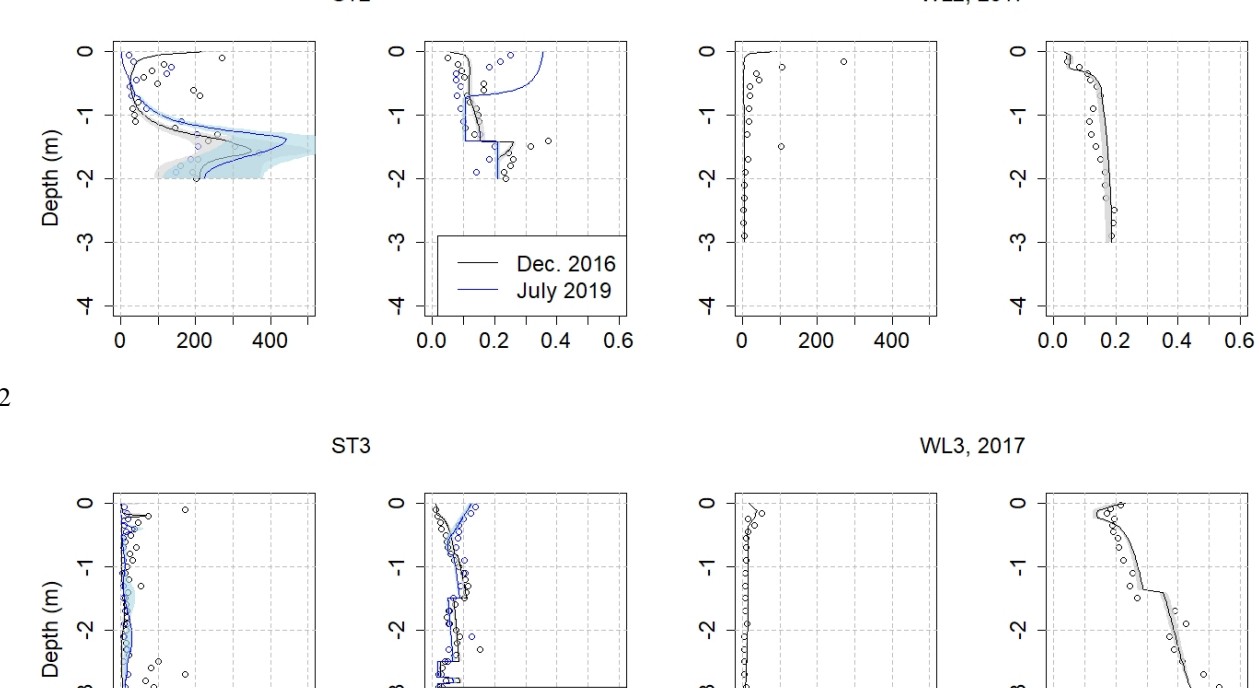



Figure 6: Measured and simulated scenarios of chloride concentration and water content for all six soil profiles. Shaded areas represent the standard deviation of 100 randomly selected model runs.


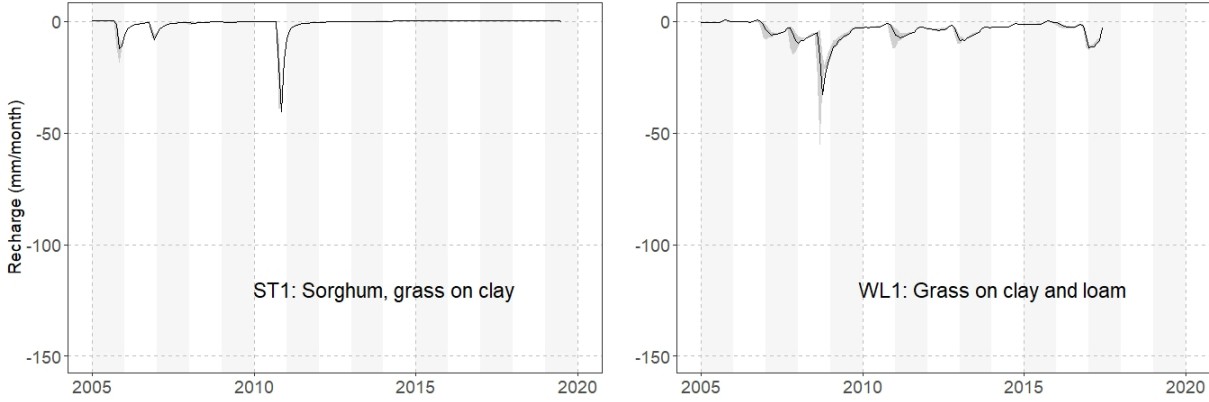


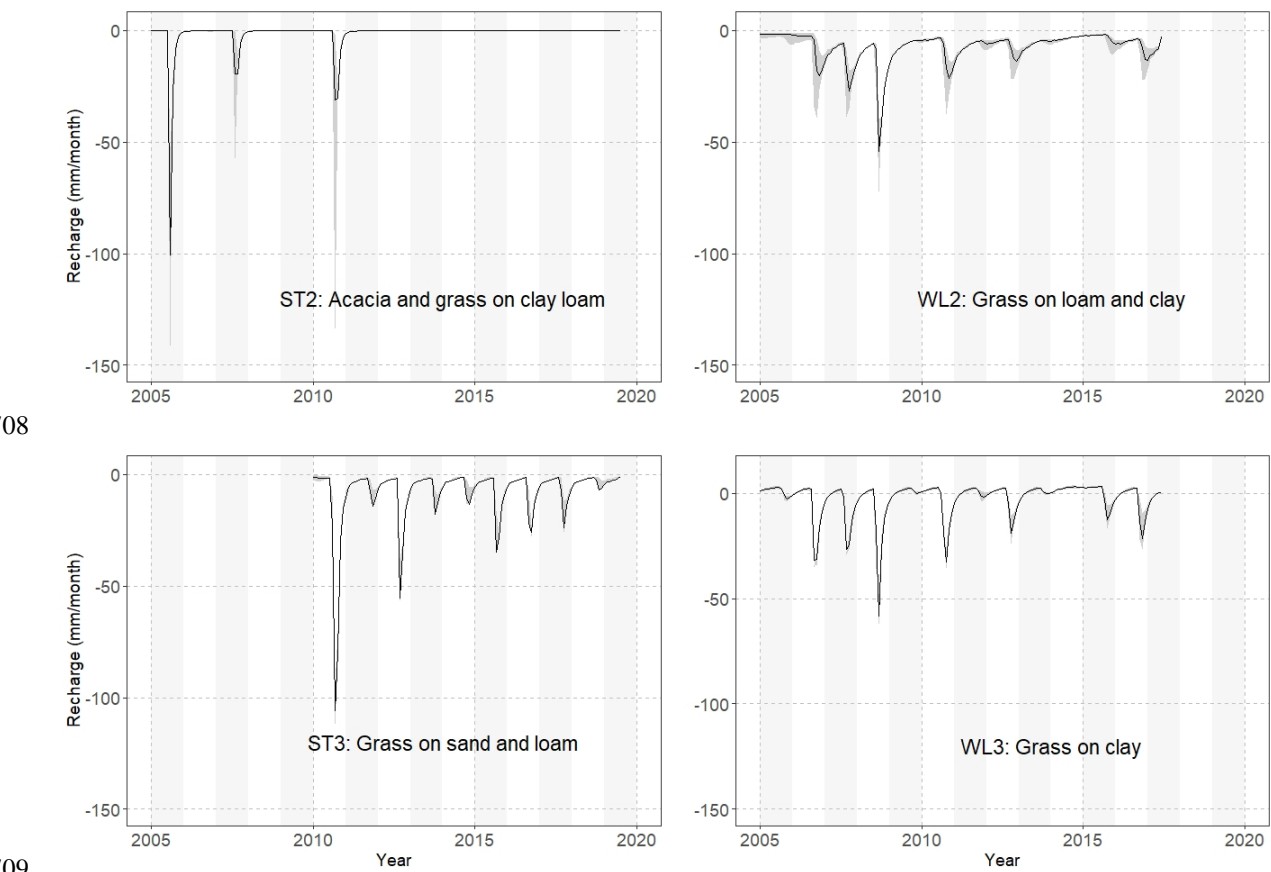



**Figure 7: Calculated groundwater recharge for all scenarios and sampling locations with indication of vegetation and soil texture.**

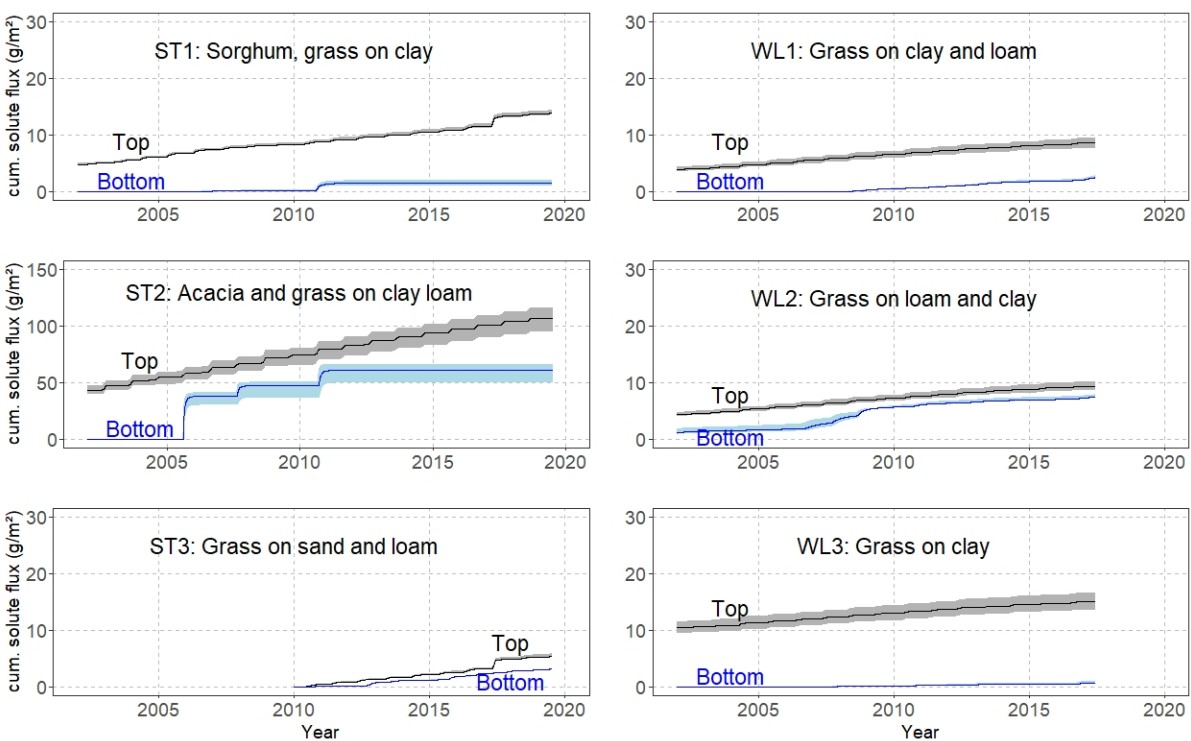


**Fig. 8: Cumulative solute flux on the upper and lower boundary of the models. The shaded areas represent the standard deviation**
**of 100 randomly selected model runs. Note the different y-axis scales between sites.**

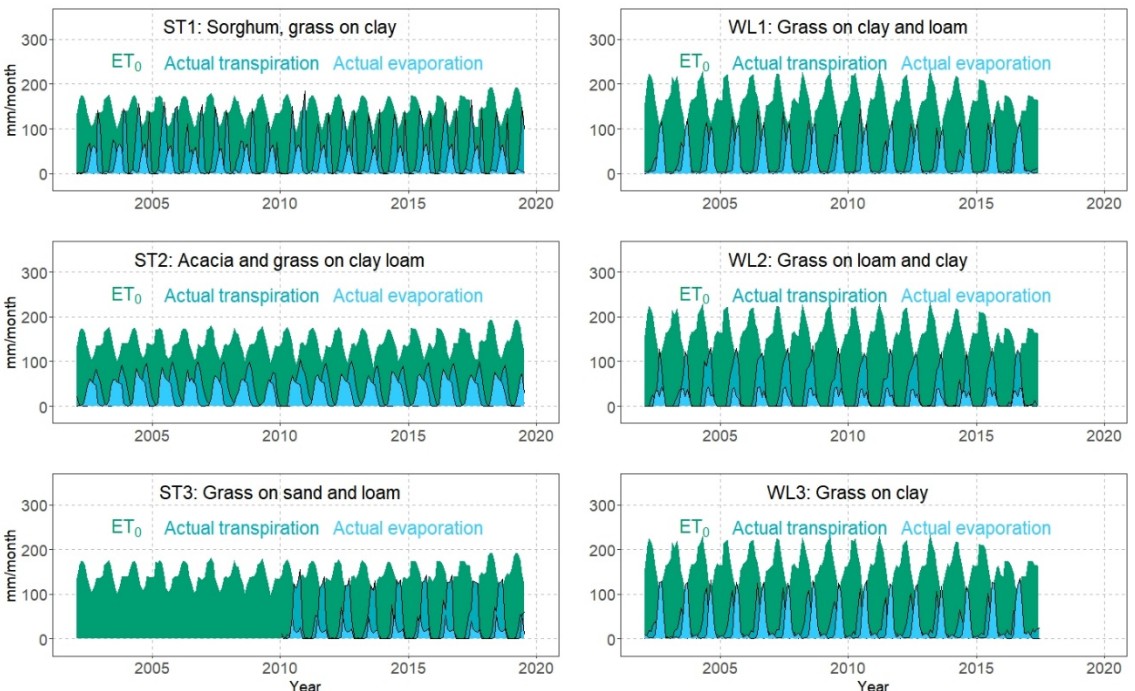


**Fig. 9: Reference evapotranspiration from the CRUTS 4 database (NCAR 2017) as well as modelled average actual evaporation and**
**transpiration of 100 randomly selected model runs.**

**Table 1: Names and geographic coordinates of the sampling locations with average depths to groundwater.**

| Name | Location | Date of Sampling | Drilling depth (m) | Longitude (°) | Latitude (°) | Elevation (m a.s.l.) | Depth to Groundwater (m) |
|---|---|---|---|---|---|---|---|
| ST1 | Gos Djarat | 07-12-2016 | 2.35 | 19.89644 | 11.02582 | 418 | 11 |
| | | 11-07-2019 | 5.0 | | | | |
| ST2 | Kach | 09-12-2016 | 2.0 | 20.07473 | 10.81649 | 396 | 16-18 |
| | Kacha | 16-07-2019 | 5.0 | | | | |
| ST3 | Gos Djarat | 11-12-2016 | 2.2 | 19.91687 | 11.00629 | 418 | 21 |
| | | 13-07-2019 | 5.0 | | | | |
| WL1 | Katoa | 01-06-2017 | 4.0 | 15.09235 | 10.82508 | 362 | 4 |
| WL2 | Loutou | 01-06-2017 | 3.0 | 15.37817 | 10.76805 | 325 | 11-12 |
| WL3 | Zina | 08-06-2017 | 3.8 | 14.97363 | 11.28858 | 304 | 3.6 |



**Table 2: Crop evapotranspiration scenarios used with the individual soil profiles.**

| Scenario | $K_{cb}$ | $K_e$ | Root depth | Profile |
|---|---|---|---|---|
| Mean | average | average | average | All profiles |
| Min | minimum | minimum | average | All profiles |
| Min-RD | minimum | minimum | minimum | WL1 |
| Mix-1 | minimum | average | average | All profiles |
| Mix-2 | average | minimum | average | ST1, WL2, WL3 |
| Mix-3 | maximum | average | average | ST3 |
| Max | maximum | maximum | average | All profiles |


**Table 3: Parametrization of water retention and unsaturated hydraulic conductivity functions according the Mualem-van**
**Genuchten model after Bayesian model calibration.**

| Location | Texture | Depth (m) | $\theta_r$ (-) | $\theta_s$ (-) | $\alpha$ (m$^{-1}$) | n (-) | $k_s$ (md$^{-1}$) |
|---|---|---|---|---|---|---|---|
| ST1 | Clay | 0-1 | 0.001 | 0.61±0.01 | 2.13±0.27 | 1.164±0.008 | 0.09±0.14 |
| | Sandy clay | 1-2.35 | 0.04 | 0.43±0.03 | 2.63±0.37 | 1.150±0.011 | 0.43±0.39 |
| ST2 | Sandy clay loam | 0-1.4 | 0.04 | 0.38±0.02 | 1.18±0.08 | 1.36±0.047 | 0.03±0.16 |
| | Clay | 1.4 -2.1 | 0.07 | 0.48±0.08 | 2.66±0.36 | 1.203±0.052 | 0.11±0.28 |
| ST3 | Loamy fine sand | 0-0.8 | 0.01 | 0.45±0.08 | 3.69±0.08 | 2.332±0.196 | 2.96±5.72 |
| | Sandy clay loam | 0.8-1.5 | 0.043 | 0.38±0.07 | 2.81±0.43 | 2.210±0.172 | 2.44±4.19 |
| | Sandy loam | 1.5-2.4 | 0.02 | 0.43±0.08 | 3.44±0.51 | 2.469±0.330 | 1.66±2.84 |
| | Loamy sand | 2.4-2.5 | 0 | 0.35±0.06 | 3.77±0.53 | 1.980±0.265 | 2.03±3.11 |
| | Sand | 2.5-2.75 | 0 | 0.34±0.04 | 3.73±0.53 | 2.730±0.372 | 5.42±8.86 |
| | Loamy sand | 2.75-2.84 | 0 | 0.35±0.06 | 3.77±0.53 | 1.980±0.265 | 2.03±3.11 |
| | Sand | 2.84-2.9 | 0 | 0.34±0.04 | 3.73±0.53 | 2.730±0.372 | 5.42±8.86 |
| WL1 | Clay | 0-0.3 | 0.065 | 0.56±0.09 | 1.37±0.19 | 1.293±0.092 | 0.17±0.26 |
| | Sandy clay | 0.3-0.9 | 0.06 | 0.44±0.07 | 2.85±0.36 | 1.416±0.125 | 0.21±0.38 |
| | Clay | 0.9-2.0 | 0.103 | 0.42±0.03 | 1.55±0.21 | 1.187±0.065 | 0.19±0.42 |
| | Sandy clay loam | 2.0-2.8 | 0.075 | 0.49±0.07 | 2.34±0.33 | 1.598±0.227 | 0.13±0.28 |
| | Sandy clay | 2.8-3.8 | 0.081 | 0.43±0.06 | 2.60±0.35 | 1.266±0.134 | 0.09±0.19 |
| | Sandy clay loam | 3.8-4.0 | 0.071 | 0.40±0.05 | 2.69±0.37 | 1.291±0.137 | 0.12±0.24 |
| WL2 | Sandy clay loam | 0-0.2 | 0.03 | 0.41±0.07 | 3.22±0.45 | 1.502±0.151 | 0.30±0.57 |
| | Sandy clay | 0.2-0.6 | 0.01 | 0.37±0.06 | 2.56±0.39 | 1.422±0.081 | 0.09±0.19 |

| | Sandy clay loam | 0.6-3.0 | 0.01 | 0.37±0.03 | 1.39±0.19 | 1.566±0.06 | 0.10±0.10 |
|---|---|---|---|---|---|---|---|
| | Sandy clay | 0-1.4 | 0.09 | 0.49±0.09 | 1.27±0.15 | 1.470±0.111 | 0.22±0.14 |
| WL3 | Clay | 1.4-3.6 | 0.105 | 0.53±0.05 | 2.03±0.29 | 1.285±0.100 | 0.17±0.36 |
| | Loamy fine sand | 3.6-3.8 | 0.056 | 0.39±0.08 | 2.90±0.45 | 1.789±0.293 | 1.23±2.40 |

**Table 4: Average root mean square error (RMSE) and related standard deviation (SD) over all scenarios for water content (Theta) and chloride concentration.**

| Location, Year | Theta ($cm^3$ $cm^{-3}$) | | | Chloride concentration (mg $l^{-1}$) | | |
|---|---|---|---|---|---|---|
| | Average observation | Average simulation | Average RMSE | Average observation | Average simulation | Average RMSE |
| ST1, 2016/2019 | 0.25/0.22 | 0.23/0.23 | 0.06/0.04 | 30/6 | 18/22 | 19/19 |
| ST2, 2016/2019 | 0.17/0.14 | 0.16/0.15 | 0.06/0.04 | 162/106 | 132/229 | 82/116 |
| ST3, 2016/2019 | 0.06/0.08 | 0.07/0.06 | 0.02/0.02 | 42/10 | 6/13 | 58/10 |
| WL1, 2017 | 0.27 | 0.27 | 0.05 | 31 | 6 | 59 |
| WL2, 2017 | 0.13 | 0.15 | 0.02 | 40 | 3 | 117 |
| WL3, 2017 | 0.31 | 0.33 | 0.04 | 12 | 13 | 9 |

**Table 5: Calculated average annual recharge, fraction of recharge on average annual precipitation, standard deviations of recharge across the time-period 2005-2019 and 2005 – 2016 for Salamat and Waza Logone, respectively.**

| Location | Average annual recharge (mm) | Fraction of average annual precipitation (%) | Standard deviation of annual recharge (mm) |
|---|---|---|---|
| ST1 | 7 | 0.9 | 17 |
| ST2 | 9 | 1 | 29 |
| ST3 | 93 | 12 | 69 |
| WL1 | 28 | 4 | 32 |
| WL2 | 54 | 8 | 46 |
| WL3 | 6 | 1 | 48 |

**Table 6: Calculated average annual evaporation and transpiration and related standard deviations of 100 randomly accepted model**
**runs.**

| Location | Average annual evaporation (mm) | Standard deviation of evaporation (mm) | Average annual transpiration (mm) | Standard deviations of transpiration (mm) | Average actual evapotranspiration (mm) |
|---|---|---|---|---|---|
| ST1 | 210 | 9 | 553 | 11 | 763 |
| ST2 | 366 | 22 | 388 | 27 | 754 |
| ST3 | 137 | 12 | 552 | 11 | 689 |
| WL1 | 344 | 20 | 317 | 23 | 661 |
| WL2 | 146 | 14 | 477 | 28 | 623 |
| WL3 | 376 | 12 | 305 | 10 | 681 |



Supplement Material
**Table S1: Precipitation chloride concentration measured in N'Djamena.**

| Sampling Date | Precipitation amount (mm) | Chloride concentration (mg l$^{-1}$)l |
|---|---|---|
| 16/06/2016 | 19 | 1.25 |
| 17/06/2016 | 17 | 0.82 |
| 26/06/2016 | 20 | 0.37 |
| 12/07/2016 | 55 | 0.13 |
| 25/07/2016 | 0.5 | 0.37 |
| 01/08/2016 | 51 | 0.29 |
| 07/08/2016 | 13.5 | 0.17 |
| 10/08/2016 | 23 | 0.22 |
| 24/08/2016 | 29 | 0.22 |
| 01/09/2016 | 36 | 0.20 |
| 24/05/2017 | 1 | 19.60 |
| 30/05/2017 | 4 | 3.40 |
| 10/06/2017 | 1 | 3.24 |
| 12/06/2017 | 12 | 0.82 |
| 21/06/2017 | 33 | 1.45 |
| 22/06/2017 | 8 | 1.12 |
| 27/06/2017 | 11 | 0.97 |
| 30/06/2017 | 6 | 0.53 |
| 05/07/2017 | 40 | 0.39 |
| 08/07/2017 | 30 | 0.17 |
| 17/07/2017 | 24 | 0.19 |
| 24/07/2017 | 65 | 0.15 |
| 09/08/2017 | 55 | 0.15 |
| 15/08/2017 | 23 | 0.28 |
| 18/08/2017 | 40 | 0.17 |
| 14/09/2017 | 29 | 0.31 |
| 25/09/2017 | 31 | 0.43 |
| 28/05/2018 | 28 | 0.91 |

| Sampling Date | Precipitation amount (mm) | Chloride concentration (mg l$^{-1}$)l |
|---|---|---|
| 01/07/2018 | 60 | 0.16 |
| 07/07/2018 | 65 | 0.11 |
| 11/07/2018 | 46 | 0.16 |
| 13/07/2018 | 43 | 0.38 |
| 30/07/2018 | 50 | 0.09 |
| 08/08/2018 | 50 | 0.13 |
| 23/08/2018 | 80 | 0.05 |
| 07/09/2018 | 30 | 0.13 |
| 28/05/2019 | 45 | 1.89 |
| 30/05/2019 | 30 | 0.37 |
| 22/07/2019 | 121 | 0.17 |
| 05/08/2019 | 29 | 0.09 |
| 09/08/2019 | 45 | 0.08 |
| 27/08/2019 | 35 | 0.25 |
| 09/09/2019 | 45 | 0.46 |
| 15/09/2019 | 43 | 0.37 |
| 01/10/2019 | 36 | 0.55 |
| 04/10/2019 | 14 | 0.46 |
| 07/10/2019 | 20 | 0.27 |
| 04/05/2020 | 19.3 | 1.35 |
| 05/06/2020 | 15.9 | 1.30 |
| 09/06/2020 | 21.4 | 0.69 |
| 15/06/2020 | 26.5 | 0.29 |
| 19/06/2020 | 12 | 0.30 |
| 26/06/2020 | 14.4 | 0.85 |
| 05/07/2020 | 26.5 | 0.21 |
| 10/07/2020 | 27.4 | 0.18 |
| 14/07/2020 | 9.4 | 0.26 |
| 22/07/2020 | 13.2 | 0.16 |
| 24/07/2020 | 16.6 | 0.12 |
| 27/07/2020 | 24.6 | 0.15 |


**Table S2: Soil chloride concentration measured in each of the Salamat profiles. GW (2016): chloride concentration in groundwater**
**and year of measurement.**

| Site | Depth interval (cm) | Soil chloride concentration (mg l$^{-1}$) | | Gravimetric water content (%) | |
|------|------|------|------|------|------|
| | | 2016 | 2019 | 2016 | 2019 |
| ST1 | 0-10 | 44.75 | 11.17 | 0.17 | 0.31 |
| | 10-20 | 42.14 | 15.05 | 0.18 | 0.18 |
| | 20-30 | 45.55 | 9.75 | 0.16 | 0.16 |
| | 30-40 | 29.40 | 7.07 | 0.28 | 0.20 |
| | 40-50 | 13.53 | 6.26 | 0.33 | 0.21 |
| | 50-60 | 32.15 | 3.82 | 0.34 | 0.23 |
| | 60-80 | 18.54 | 2.78 | 0.33 | 0.24 |
| | 80-100 | 12.26 | 2.64 | 0.28 | 0.24 |
| | 100-110 | 14.55 | 2.12 | 0.29 | 0.25 |
| | 110-120 | 24.89 | | 0.29 | |
| | 120-130 | 29.28 | 1.72 | 0.28 | 0.23 |
| | 130-140 | 18.57 | | 0.27 | |
| | 140-150 | 24.00 | 1.74 | 0.25 | 0.23 |
| | 150-160 | 55.14 | | 0.27 | |
| | 160-170 | 51.84 | 1.91 | 0.22 | 0.23 |
| | 170-180 | 24.65 | | 0.25 | |
| | 180-190 | 40.14 | 2.25 | 0.23 | 0.20 |
| | 190-200 | 28.30 | | 0.22 | |
| | 200-210 | 42.37 | 15.5 | 0.22 | 0.20 |
| | 210-220 | 10.07 | | 0.21 | |
| | 220-230 | 26.71 | 1.47 | 0.21 | 0.18 |
| | 230-240 | 24.25 | | 0.22 | |
| | 240-260 | | 1.75 | | 0.16 |
| | 260-280 | | 1.42 | | 0.17 |
| | 280-300 | | 1.26 | | 0.16 |
| | 300-320 | | 1.10 | | 0.16 |
| | 320-340 | | 1.76 | | 0.16 |
| | 340-360 | | 2.23 | | 0.14 |

| Site | Depth interval (cm) | Soil chloride concentration (mg l⁻¹) | | Gravimetric water content (%) | |
|---|---|---|---|---|---|
| | | 2016 | 2019 | 2016 | 2019 |
| | 360-380 | | 1.14 | | 0.16 |
| | 380-400 | | 0.99 | | 0.16 |
| | 400-420 | | 1.41 | | 0.16 |
| | 420-440 | | 1.35 | | 0.15 |
| | 440-460 | | 1.38 | | 0.16 |
| | 460-480 | | 1.08 | | 0.17 |
| | 480-500 | | 1.02 | | 0.18 |
| GW (2019) | 1,100 | 0.34 | | | |
| ST2 | 0-10 | 269.66 | 21.07 | 0.03 | 0.25 |
| | 10-20 | 115.31 | 34.16 | 0.04 | 0.22 |
| | 20-30 | 82.18 | 134.77 | 0.05 | 0.18 |
| | 30-40 | 60.80 | 121.60 | 0.06 | 0.07 |
| | 40-50 | 98.79 | 40.65 | 0.09 | 0.08 |
| | 50-60 | 194.90 | 24.57 | 0.09 | 0.09 |
| | 60-70 | 210.85 | 28.27 | 0.06 | 0.08 |
| | 70-80 | 46.03 | | 0.06 | |
| | 80-90 | 30.47 | 69.27 | 0.08 | 0.09 |
| | 90-100 | 36.89 | | 0.08 | |
| | 100-110 | 37.52 | 160.93 | 0.08 | 0.10 |
| | 110-120 | 144.63 | | 0.05 | |
| | 120-130 | 258.28 | 202.94 | 0.07 | 0.15 |
| | 130-140 | 233.81 | | 0.17 | |
| | 140-150 | 304.90 | 206.61 | 0.16 | 0.20 |
| | 150-160 | 368.07 | | 0.12 | |
| | 160-170 | 207.07 | 186.64 | 0.12 | 0.18 |
| | 170-180 | 158.34 | | 0.13 | |
| | 180-190 | 191.84 | 146.81 | 0.12 | 0.14 |
| | 190-200 | 201.77 | | 0.12 | |
| | 200-220 | | 117.61 | | 0.14 |
| | 220-240 | | 82.31 | | 0.14 |

| Site | Depth interval (cm) | Soil chloride concentration (mg l$^{-1}$) | | Gravimetric water content (%) | |
|---|---|---|---|---|---|
| | | 2016 | 2019 | 2016 | 2019 |
| | 240-260 | | 65.10 | | 0.13 |
| | 260-280 | | 48.25 | | 0.13 |
| | 280-300 | | 32.34 | | 0.12 |
| | 300-320 | | 25.89 | | 0.12 |
| | 320-340 | | 27.25 | | 0.12 |
| | 340-360 | | 26.81 | | 0.12 |
| | 360-380 | | 41.14 | | 0.13 |
| | 380-400 | | 49.37 | | 0.14 |
| | 400-420 | | 53.52 | | 0.14 |
| | 420-440 | | 56.65 | | 0.14 |
| | 440-460 | | 55.76 | | 0.10 |
| | 460-480 | | 101.40 | | 0.01 |
| | 480-500 | | 149.20 | | 0.01 |
| GW (2016) | 1,700 | 1.39 | | | |
| ST3 | 0-10 | 172.79 | 9.27 | 0.01 | 0.09 |
| | 10-20 | 73.81 | 15.44 | 0.02 | 0.08 |
| | 20-30 | 45.54 | 18.40 | 0.02 | 0.07 |
| | 30-40 | 37.08 | 8.65 | 0.03 | 0.06 |
| | 40-50 | 25.53 | 13.64 | 0.04 | 0.05 |
| | 50-60 | 10.71 | 6.38 | 0.08 | 0.05 |
| | 60-70 | 41.26 | | 0.05 | |
| | 70-80 | 23.95 | 7.62 | 0.06 | 0.05 |
| | 80-90 | 30.15 | | 0.07 | |
| | 90-100 | 15.67 | 6.12 | 0.08 | 0.07 |
| | 100-110 | 11.67 | | 0.09 | |
| | 110-120 | 21.20 | 4.06 | 0.09 | 0.07 |
| | 120-130 | 54.53 | | 0.10 | |
| | 130-140 | 19.90 | 6.48 | 0.10 | 0.06 |
| | 140-150 | 12.27 | | 0.10 | |
| | 150-160 | 13.14 | 7.03 | 0.07 | 0.04 |

| Site | Depth interval (cm) | Soil chloride concentration (mg l$^{-1}$) | | Gravimetric water content (%) | |
|---|---|---|---|---|---|
| | | 2016 | 2019 | 2016 | 2019 |
| | 160-170 | 16.54 | | 0.05 | |
| | 170-180 | 15.62 | 11.41 | 0.04 | 0.04 |
| | 180-190 | 15.96 | | 0.05 | |
| | 190-200 | 14.55 | 10.19 | 0.07 | 0.03 |
| | 200-210 | 7.40 | | 0.08 | |
| | 210-220 | 14.77 | 8.93 | 0.10 | 0.08 |
| | 220-230 | 12.77 | | 0.12 | |
| | 230-240 | 22.53 | 15.25 | 0.07 | 0.03 |
| | 240-250 | 101.22 | | 0.03 | |
| | 250-260 | 80.82 | 7.08 | 0.03 | 0.03 |
| | 260-270 | 171.44 | | 0.02 | |
| | 270-280 | 66.03 | 18.41 | 0.03 | 0.02 |
| | 280-300 | 88.52 | 11.64 | 0.02 | 0.02 |
| | 300-320 | | 28.13 | | 0.02 |
| | 320-340 | | 24.83 | | 0.01 |
| | 340-360 | | 37.71 | | 0.01 |
| | 360-380 | | 30.59 | | 0.02 |
| | 380-400 | | 29.21 | | 0.02 |
| | 400-420 | | 32.63 | | 0.02 |
| | 420-440 | | 36.39 | | 0.02 |
| | 440-460 | | | | |
| | 460-480 | | 39.34 | | 0.02 |
| | 480-500 | | 39.51 | | 0.02 |
| GW (2016) | 2,100 | 4.10 | | | |






**Table S3: Soil chloride concentration measured in each of the Waza Logone profiles in 2017. GW (2016): chloride concentration in**
**groundwater and year of measurement.**

| Site | Depth interval (cm) | Soil chloride concentration (mg l$^{-1}$) | Gravimetric water content (%) |
|---|---|---|---|
| | 0-10 | 246.74 | 0.12 |
| | 10-20 | 142.34 | 0.13 |
| | 20-30 | 39.42 | 0.12 |
| | 30-40 | 17.99 | 0.11 |
| | 40-50 | 18.03 | 0.07 |
| | 50-60 | 26.61 | 0.08 |
| | 60-70 | 9.44 | 0.11 |
| | 70-80 | 4.36 | 0.13 |
| | 80-90 | 8.31 | 0.15 |
| | 90-100 | 11.22 | 0.18 |
| | 100-120 | 7.33 | 0.16 |
| | 120-140 | 4.26 | 0.15 |
| WL1 | 140-160 | 3.65 | 0.16 |
| | 160-180 | 12.24 | 0.18 |
| | 180-200 | 3.11 | 0.19 |
| | 200-220 | 3.00 | 0.14 |
| | 220-240 | 7.18 | 0.11 |
| | 240-260 | 22.38 | 0.15 |
| | 260-280 | 18.40 | 0.14 |
| | 280-300 | 17.46 | 0.16 |
| | 300-320 | 16.14 | 0.17 |
| | 320-340 | 46.94 | 0.18 |
| | 340-360 | 24.16 | 0.17 |
| | 360-380 | 75.87 | 0.15 |
| | 380-400 | 6.86 | 0.15 |
| GW (2017) | 400 | 0.23 | |
| | 0-10 | 446.21 | 0.03 |
| WL2 | 10-20 | 269.74 | 0.02 |
| | 20-30 | 106.04 | 0.04 |
| | 30-40 | 35.22 | 0.06 |

| Site | Depth interval (cm) | Soil chloride concentration (mg l$^{-1}$) | Gravimetric water content (%) |
|---|---|---|---|
| | 40-50 | 44.11 | 0.06 |
| | 50-60 | 19.47 | 0.07 |
| | 60-80 | 18.40 | 0.08 |
| | 80-100 | 17.55 | 0.07 |
| | 100-120 | 15.36 | 0.06 |
| | 120-140 | 12.12 | 0.07 |
| | 140-160 | 101.76 | 0.07 |
| | 160-180 | 15.07 | 0.08 |
| | 180-200 | 5.86 | 0.09 |
| | 200-220 | 4.90 | 0.09 |
| | 220-240 | 3.85 | 0.09 |
| | 240-260 | 2.69 | 0.09 |
| | 260-280 | 2.73 | 0.10 |
| | 280-300 | 4.87 | 0.10 |
| GW (2014) | 1,200 | 0.90 | |
| | 0-10 | 157.35 | 0.06 |
| | 10-20 | 50.59 | 0.09 |
| | 20-30 | 13.32 | 0.11 |
| | 30-40 | 32.41 | 0.11 |
| | 40-50 | 14.77 | 0.11 |
| | 50-60 | 9.02 | 0.12 |
| | 60-80 | 8.85 | 0.12 |
| | 80-100 | 10.27 | 0.13 |
| WL3 | 100-120 | 6.84 | 0.13 |
| | 120-140 | 6.28 | 0.13 |
| | 140-160 | 5.59 | 0.15 |
| | 160-180 | 7.63 | 0.20 |
| | 180-200 | 11.47 | 0.21 |
| | 200-220 | 4.83 | 0.19 |
| | 220-240 | 5.30 | 0.19 |
| | 240-260 | 4.74 | 0.21 |

| Site | Depth interval (cm) | Soil chloride concentration (mg l$^{-1}$) | Gravimetric water content (%) |
|------|------|------|------|
| | 260-280 | 4.98 | 0.22 |
| | 280-300 | 9.54 | 0.23 |
| | 300-320 | 10.24 | 0.21 |
| | 320-340 | 9.75 | 0.22 |
| | 340-360 | 9.25 | 0.23 |
| | 360-380 | 21.78 | 0.17 |
| GW (2017) | 360 | 1.51 | |


**Table S4: Site-specific estimated monthly variation of ground cover including grass, crops, and flooding periods with ranges of**
**monthly basal crop coefficient (Kcb), soil water evaporation coefficient (Ke), and root depth used in the scenarios.**

| Location | Month | Vegetation | Kcb | Ke | Root depth (m) |
|------|------|------|------|------|------|
| ST1 | Jan | Sorghum | 1.01 – 0.86 | 0.12 | 1.5 – 2.5 |
| | Feb | Sorghum | 0.35 – 0.35 | 0.48 | 1.5 – 2.5 |
| | Mar | Bare soil | 0.0 | 0.47 – 0.68 | 0.0 |
| | Apr | Bare soil | 0.0 | 0.27 – 0.41 | 0.0 |
| | May | Grass | 0.6 – 0.4 | 0.1 – 0.12 | 0.1 – 0.5 |
| | June | Grass | 0.85 – 0.6 | 0.05 | 0.2 – 0.7 |
| | July | Grass | 1.03 – 0.83 | 0.12 | 0.2– 0.7 |
| | Aug | Flooded | 0.3 – 0.2 | 0.9 – 1.0 | 0.2 – 0.7 |
| | Sep | Flooded | 0.0 | 1.08 | 0.0 |
| | Oct | Flooded | 0.0 | 1.08 | 0.0 |
| | Nov | Sorghum | 0.2 – 0.1 | 0.66 | 0.5 – 1.5 |
| | Dec | Sorghum | 1.01 – 0.86 | 0.12 | 1.5 – 2.5 |
| ST2 | Jan | Tree (Acacia) | 0.5 – 0.8 | 0.2 – 0.34 | |
| | Feb | Tree | 0.2 – 0.6 | 0.17 – 0.39 | |
| | Mar | Tree | 0.1 – 0.3 | 0.07 – 0.37 | |
| | Apr | Tree | 0.1 – 0.3 | 0.02 – 0.17 | Time invariant |
| | May | Tree, Grass | 0.1 – 0.3 | 0.5 | root distribution |
| | June | Tree, Grass | 0.3 – 0.4 | 0.5 | |
| | July | Tree, Grass | 0.3 – 0.4 | 0.5 | |
| | Aug | Tree, flooded | 0.6 – 0.8 | 0.25 | |

| Location | Month | Vegetation | Kcb | Ke | Root depth (m) |
|---|---|---|---|---|---|
| | Sep | Tree, flooded | 0.7 – 1.05 | 0.25 | |
| | Oct | Tree, flooded | 0.7 – 1.05 | 0.25 | |
| | Nov | Tree, Grass | 0.7 – 0.9 | 0.25 | |
| | Dec | Tree | 0.5 – 0.8 | 0.32 – 0.37 | |
| ST3 | Jan | Grass | 0.05 – 0.1 | 0.13 – 0.39 | 0.1 |
| | Feb | Dry Grass | 0.0 – 0.05 | 0.04 – 0.13 | 0.1 |
| | Mar | Dry Grass | 0.0 – 0.05 | 0.0 | 0.1 – 0.2 |
| | Apr | Grass | 0.1 – 0.3 | 0.0 | 0.2 – 0.3 |
| | May | Green grass | 0.6 | 0.0 | 0.2 – 0.3 |
| | June | Green grass | 0.65 – 0.85 | 0.34 – 0.54 | 0.2 – 0.4 |
| | July | Green grass | 0.73 – 1.03 | 0.16 - 0.24 | 0.2 – 0.4 |
| | Aug | Green grass | 0.83 – 1.03 | 0.09 - 0.22 | 0.2 – 0.4 |
| | Sep | Green grass | 0.83 – 0.98 | 0.14 - 0.22 | 0.2 – 0.4 |
| | Oct | Green grass | 0.78 – 0.93 | 0.19 - 0.22 | 0.2 – 0.3 |
| | Nov | Green grass | 0.4 – 0.6 | 0.51 | 0.1 – 0.3 |
| | Dec | Grass | 0.05 – 0.1 | 0.4 – 0.67 | 0.1 |
| WL1 | Jan | Bare soil | 0.0 – 0.1 | 0.29 – 0.63 | 0.0 – 0.2 |
| | Feb | Bare soil | 0.0 – 0.1 | 0.16 – 0.42 | 0.0 – 0.2 |
| | Mar | Bare soil | 0.10 | 0.07 – 0.25 | 0.05 – 0.3 |
| | Apr | Grass | 0.30 – 0.4 | 0.03 – 0.1 | 0.1 – 0.3 |
| | May | Grass | 0.40 | 0.02 – 0.08 | 0.1 – 0.5 |
| | June | Grass | 0.70 – 0.9 | 0.01 – 0.02 | 0.1 – 0.5 |
| | July | Grass | 0.89 – 1.09 | 0.13 | 0.1 – 0.5 |
| | Aug | Flooded | 0.1 – 0.2 | 0.92 - 1.01 | 0.1 – 0.4 |
| | Sep | Flooded | 0.1 – 0.2 | 0.92 - 1.01 | 0.0 – 0.1 |
| | Oct | Flooded | 0.1 – 0.2 | 0.92 - 1.01 | 0.0 – 0.1 |
| | Nov | Grass | 0.2 – 0.5 | 0.58 – 0.62 | 0.05 – 0.2 |
| | Dec | Grass | 0.2 – 0.4 | 0.37 – 0.57 | 0.05 – 0.2 |
| WL2 | Jan | Bare soil | 0.0 – 0.1 | 0.11 – 0.57 | 0.0 – 0.1 |
| | Feb | Bare soil | 0.0 – 0.1 | 0.03 – 0.24 | 0.0 – 0.1 |
| | Mar | Bare soil | 0.1 – 0.2 | 0.0 – 0.07 | 0.05 – 0.1 |
| | Apr | Grass | 0.2 – 0.3 | 0.0 – 0.02 | 0.05 – 0.2 |

| Location | Month | Vegetation | Kcb | Ke | Root depth (m) |
|----------|-------|------------|-----|-----|----------------|
| | May | Grass | 0.4 – 0.4 | 0.0 – 0.01 | 0.15 – 0.3 |
| | June | Grass | 0.45 – 0.6 | 0.0 | 0.15 – 0.3 |
| | July | Grass | 0.5 – 0.7 | 0.55 - 0.75 | 0.15 – 0.3 |
| | Aug | Grass | 0.45 – 1.0 | 0.12 - 0.22 | 0.1 – 0.3 |
| | Sep | Flooded | 0.3 – 1.0 | 0.12 - 0.22 | 0.1 – 0.3 |
| | Oct | Grass | 0.2 – 0.7 | 0.42 - 0.67 | 0.15 – 0.2 |
| | Nov | Grass | 0.1 – 0.2 | 0.77 | 0.15 – 0.2 |
| | Dec | Grass | 0.0 – 0.1 | 0.32 – 0.9 | 0.05 – 0.1 |
| | Jan | Bare soil | 0.0 – 0.1 | 0.28 – 0.6 | 0.1 – 0.2 |
| | Feb | Bare soil | 0.0 – 0.1 | 0.15 – 0.4 | 0.1 – 0.2 |
| | Mar | Bare soil | 0.1 – 0.1 | 0.06 – 0.23 | 0.1 – 0.5 |
| | Apr | Grass | 0.2 – 0.4 | 0.03 – 0.09 | 0.2 – 0.5 |
| | May | Grass | 0.4 | 0.01 – 0.08 | 0.2 – 0.6 |
| WL3 | June | Grass | 0.4 – 0.9 | 0.01 - 0.01 | 0.2 – 0.6 |
| | July | Grass | 0.79 – 1.09 | 0.13 | 0.2 – 0.6 |
| | Aug | Flooded | 0.1 – 0.2 | 0.92 - 1.01 | 0.2 – 0.6 |
| | Sep | Flooded | 0.0 – 0.1 | 1.01 | 0.2 – 0.6 |
| | Oct | Flooded | 0.0 – 0.1 | 1.01 | 0.2 – 0.5 |
| | Nov | Grass | 0.2 – 0.5 | 0.58 | 0.1 – 0.5 |
| | Dec | Grass | 0.1 – 0.4 | 0.1 - 0.41 | 0.1 – 0.2 |



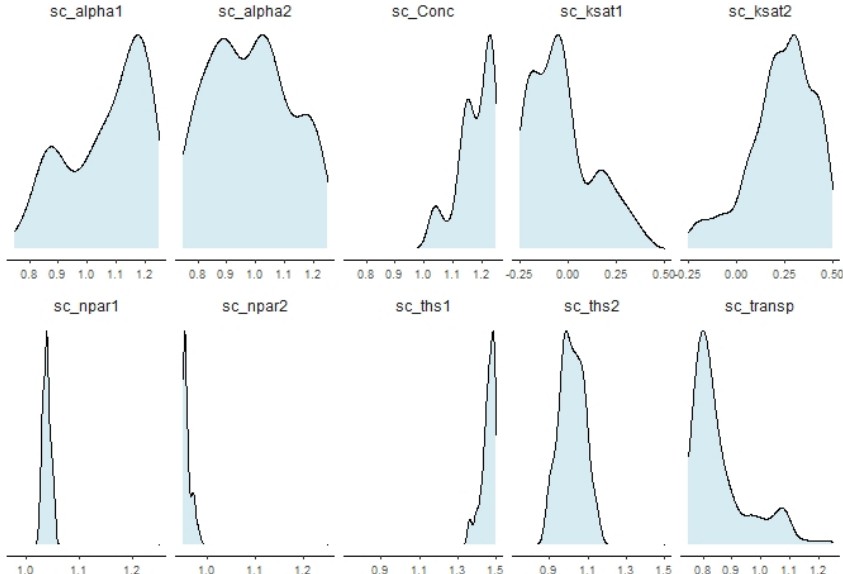


**Fig. S1: posterior density distributions of the scaling factors used in the calibration of model ST1. Numbers indicate the individual model layers. Range of x-axes corresponds to prior distribution (parameter alpha of the Mualem-van Genuchten equation: sc_alpha1. sc_alpha2: alpha; sc_Conc: input chloride concentration; sc_ksat1. sc_ksat2: saturated hydraulic conductivity; sc_npart1. sc_npart2: parameter n of the Mualem-van Genuchten equation ; sc_ths1. sc_ths2: saturated water content; sc_transp: transpiration fraction in the evapotranspiration; 1: upper layer; 2: lower layer).**

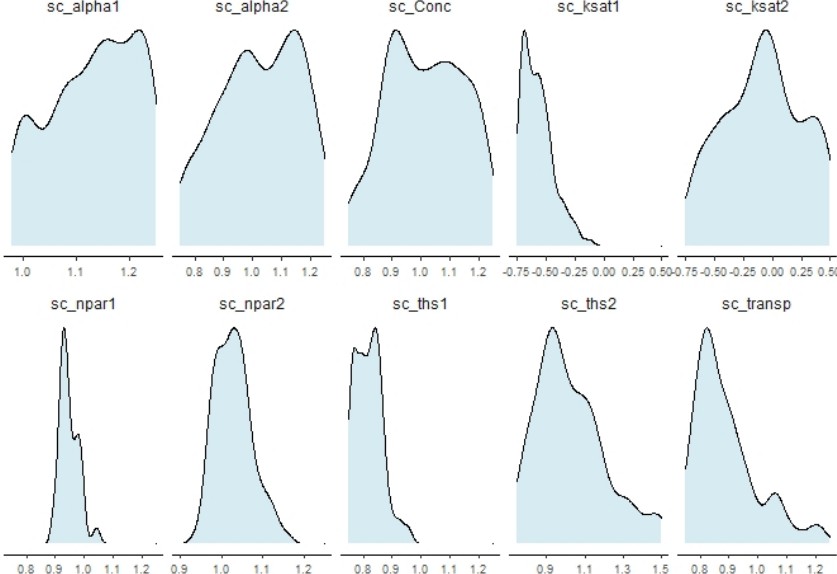


**Fig. S2: posterior density distributions of the scaling factors used in the calibration of model ST2. Numbers indicate the individual model layers. Range of x-axes corresponds to prior distribution.**

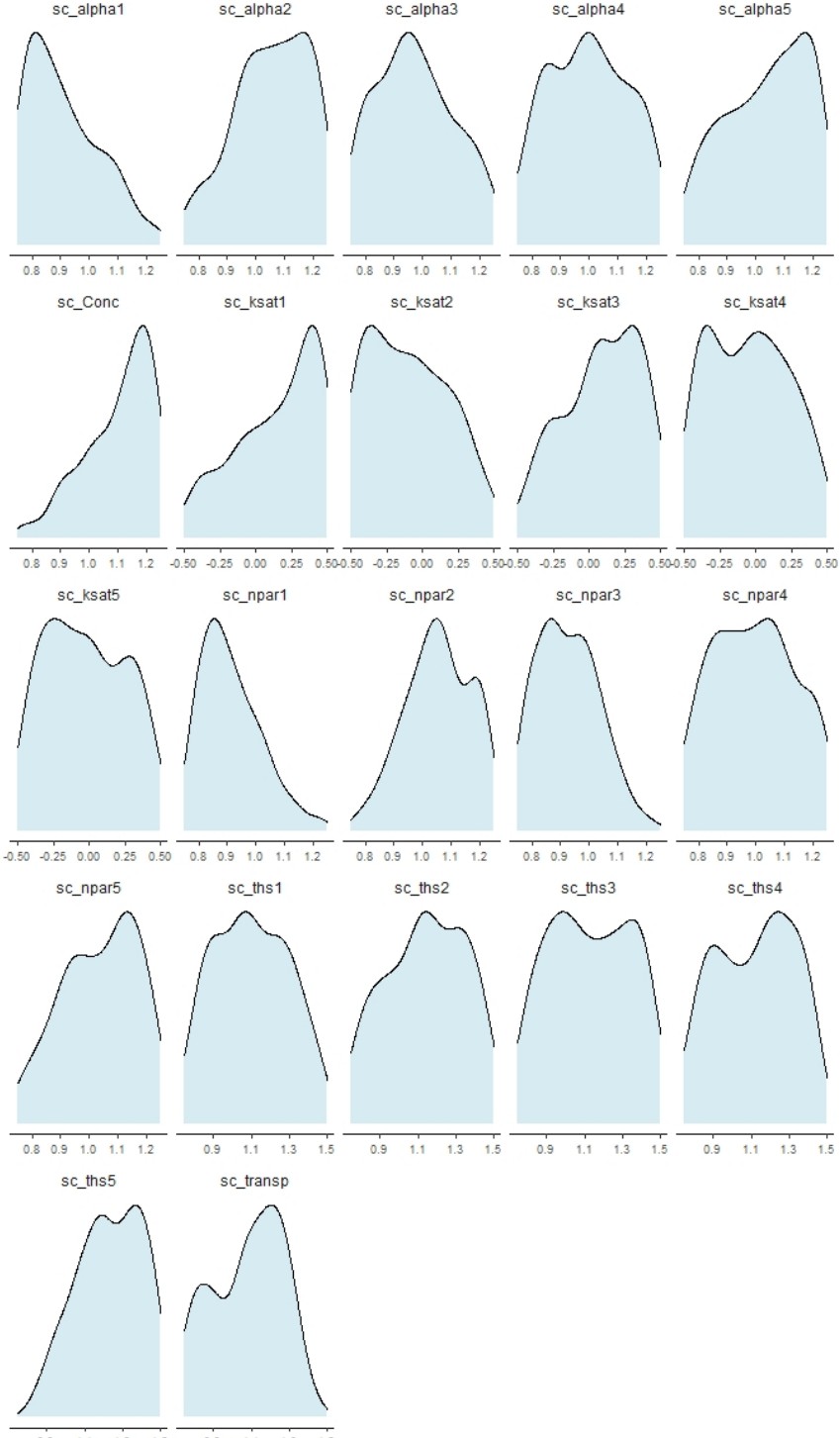


Fig. S3: posterior density distributions of the scaling factors used in the calibration of model ST3. Numbers indicate the individual
model layers. Range of x-axes corresponds to prior distribution.

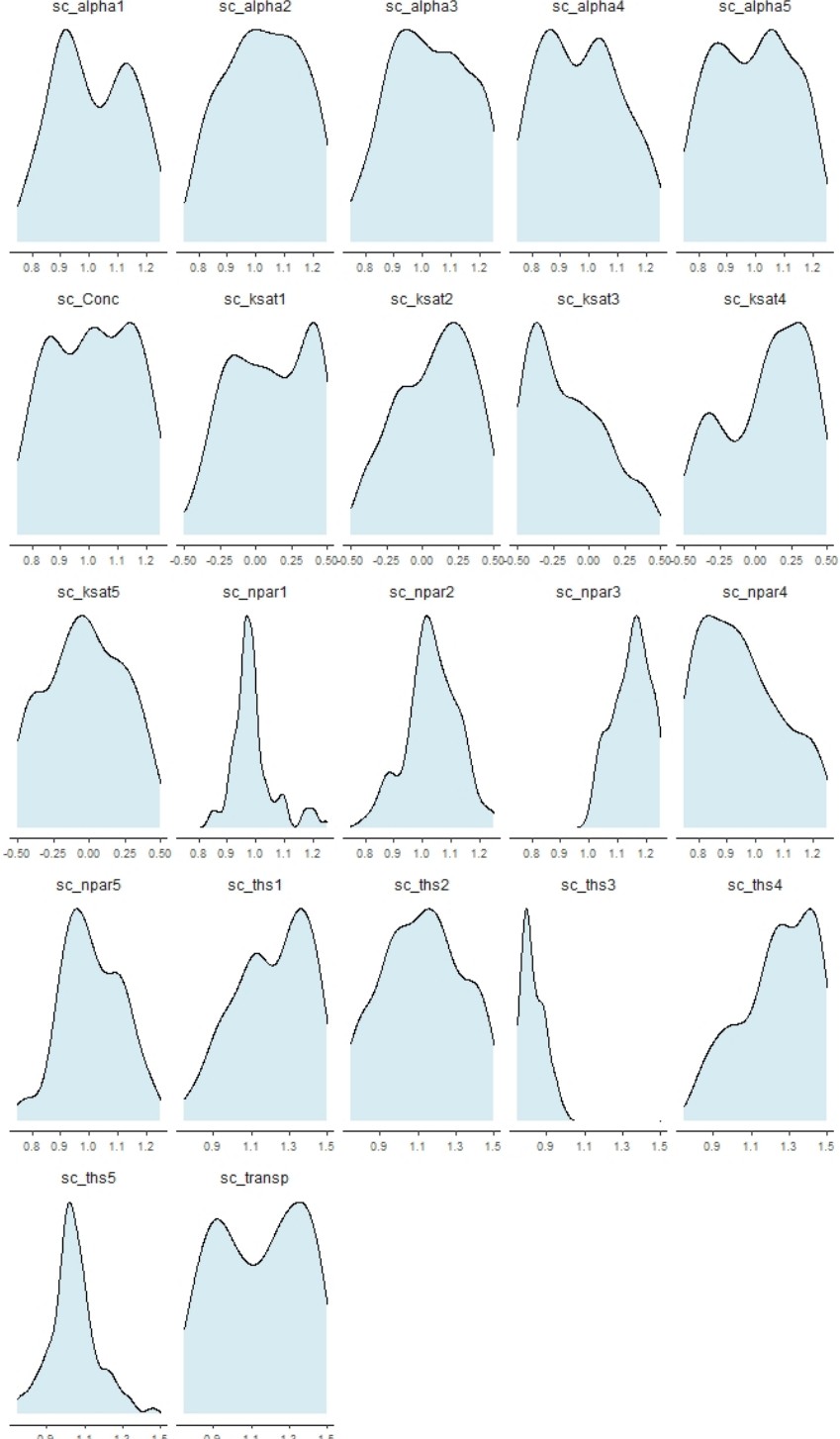


Fig. S4: posterior density distributions of the scaling factors used in the calibration of model WL1. Numbers indicate the individual model layers. Range of x-axes corresponds to prior distribution.

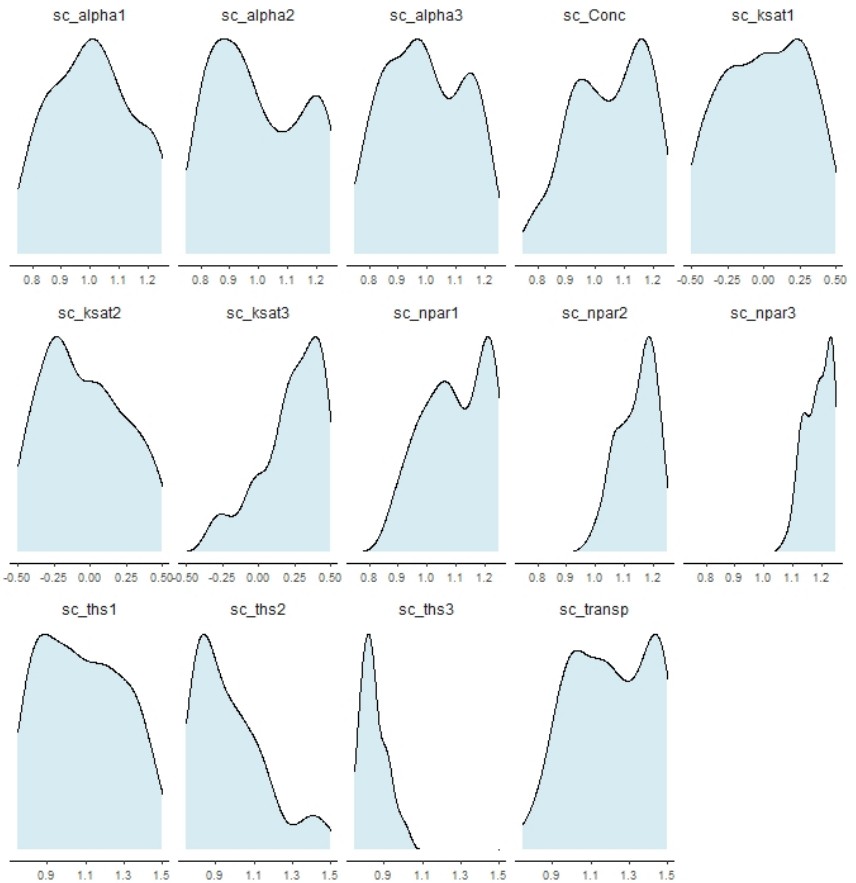


**Fig. S5: posterior density distributions of the scaling factors used in the calibration of model WL2. Numbers indicate the individual model layers. Range of x-axes corresponds to prior distribution.**

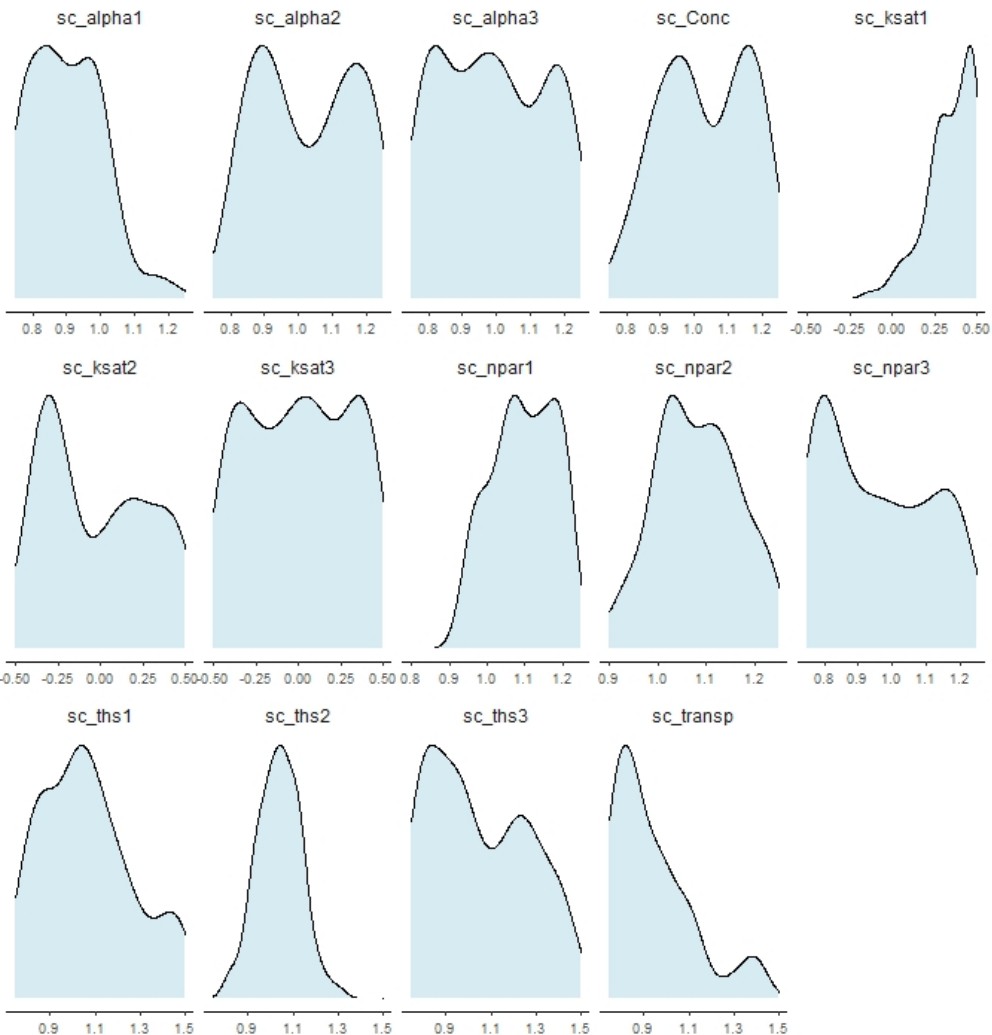


Fig. S6: posterior density distributions of the scaling factors used in the calibration of model WL3. Numbers indicate the individual
model layers. Range of x-axes corresponds to prior distribution.