# Peer review of "Modelling groundwater recharge, actual evaporation and transpiration in semi-arid sites of the Lake Chad Basin: The role of soil and vegetation on groundwater recharge"

_Hydrology and Earth System Sciences, 2022_

## Author Comment (AC1)

**Review Report**

**Title:** Modelling groundwater recharge, actual evaporation and transpiration in semi-arid sites of the Lake Chad Basin: The role of soil and vegetation on groundwater recharge

**Author(s):** Christoph Neukum et al.

**MS No.:** hess-2022-319

**General Comments**

In this research, the estimation of Evapotranspiration (ET) and groundwater recharge in Chad Lake Basin (CLB) has been done using unsaturated zone studies and modelling approach. In this regards, the authors collected soil samples from six boreholes and measured grain sizes (soil texture), water content and chloride concentrations. In addition they used climatic data (precipitation amount and Cl content) and vegetation cover characteristics to calculate the ET by a dual-crop coefficient method (Kc=Kcb+Ke). Hydrus-1D software was used to model unsaturated flow and transport and then to simulate the groundwater recharge and separated evaporation and transpiration values.

In my opinion, the structure of the manuscript is fairly appropriate as it generally represents a good example of unsaturated zone modelling. Although the results are highly site-specific, the collected data and modelling approach could be interesting for the readers of the HESS Journal.

**Specific comments**

More explanations are needed about the criteria for selecting the sites (soil profiles) in LCB as they are so close and limited. Regarding the extensive area of the LCB, are the selected sites representative of the region? Is it possible for upscaling the results from these limited sites to the whole LCB? What is the recommended strategy for upscaling results in LCB as a whole?

Selection of sites was limited mainly by accessibility and project's goals. At the time of sampling, the project concentrated in study cases in Waza Logone and Salamat.
The types of soils we have worked with (sand, loam, clay and their combinations) are the most common in the LCB. However, due to the extension of the LCB, we surely do not cover all existent soils.
We do not intend to extrapolate our values to the whole basin. We are very much aware that this would be an impossible work. What we want to show is that, using a generalised model, it is possible to determine recharge rates in areas with low accessibility and lack of data.

Why the bulk densities were not measured in the field? (Line 169)

Because of the difficulties handling the samples and sending them to Germany for measurement. We are aware of the limited accuracy of available methods, which increases with sampling depth (Al-Shammary et al. 2018[1]).
* * *
[1] https://doi.org/10.1016/S1002-0160(18)60034-7

Regarding the uncertainties inherited with the modeling approaches especially in unsaturated zone with more limited and unknown data, how do you confirm the modeling results on simulated ET and groundwater recharge values?

Results of evapotranspiration were not confirmed. However, the estimated values of the soil model as well as the calculated results are within plausible ranges. Our recharge values are in accordance with other studies, e.g. Bouchez et al., 2019[2].

We confirm our estimated recharge values with those published for the same area (Lines 451-454). We are not able to confirm them by other methods (groundwater level variation, lysimeter), because they do not exist in our study area

In the case of groundwater recharge you need to verify the modelling results by presenting the groundwater hydrographs and show any consistency between the recharge time series and water table fluctuations and then confirm the reliability of the method and results.

We agree with you, but these data are not available in the LCB, at least not in our study regions. This is the challenge of working in data scarce areas and one important motivation of this study.

Please explain in the text, why you used the both flow and transport modelling for estimation of ET and groundwater? Regarding the higher uncertainties in transport models, the basis for implementing transport model needs to be clarified as it was possible to estimate both ET and groundwater recharge by a flow model, only.

Our model was calibrated using measured values of chloride and water contend with depth. Thus, transport model was necessary. This is already explained in chapter 3.

**Technical corrections**

Line 1: In the title "actual evaporation and transpiration" is better to be replaced as "actual evapo-transpiration".

We prefer to leave as it is since we calculate both physical quantities separately

Line 74: check the English "Pedotransfer functions (PTF) bridge available and needed data and are frequently used to".

Corrected as: Pedotransfer functions (PTF) bridge available and needed data. They are frequently used to…

Line 108-109: The sentence is redundant, better to be deleted.

Done

Line 121: ST1 has not shown on Fig.1.
* * *
[2] Bouchez, C., Deschamps P., Goncalves J., Hamelin, B., Nour, A.M., Vallet-Coulomb C., and Sylvestre, F: Water transit time and active recharge in the Sahel inferred by bomb-produced 36Cl. Nature, scientific reports, 9: 7465, (2019).

Sorry! St1, St2, and ST3 were shown as S1, S2, and S3 in the map. The map has been corrected

Line 331: The figure caption (Fig. S1.) needs more clarification. You need to explain the abbreviations.

Done

---

## Author Comment (AC2)

This paper aims at evaluating groundwater recharge in a semi-arid area, the Lake Chad Basin, which is an important and difficult task. The authors use soil water contents and chloride concentrations measured in the unsaturated zone and a 1D-model to simulate water flows and chloride contents. The approach Is repeated at six locations over the catchment and allows for the estimation of ET and groundwater recharge along 15 years. This work shows that the interannual variability of groundwater recharge is first controlled by soil texture and vegetation, with lower recharge variability in coarse soils with grass cover. It also nicely shows different chloride retention in soils.

The paper is an interesting case study of unsaturated zone flux modelling but some improvements and clarifications in the manuscript are required. Following are my comments :

1.      In the abstract, the authors say that it is a generalized approach. However in the example given here, it actually seems very localized and site-specific. The results are different between each soil, which suggest that we would need a large number of soil profiles to estimate recharge over the catchment. Are the results obtained generalizable? Are the soils and vegetation types studied here covering all expected soils and vegetation types of the LCB? How do the authors extrapolate the local recharge estimation to an average recharge rate?

The sentence reads "A simple, generalized approach, which requires only limited data…". We describe a generalized approach; we do not say that the results are generalizable.
The types of soils we have worked with (sand, loam, clay and their combinations) are the most common in the LCB. However, due to the extension of the LCB, we surely do not cover all existent soils. Concerning vegetation, acacia and grass are the most widespread natural vegetation throughout the LCB, whereas sorghum is the most commonly planted corn. Cotton, which is also planted, is only locally produced and generally using irrigation. Mango trees can be found along the Chari and Logone rivers, but are not representative for the whole LCB.
We do not intend to extrapolate our values to the whole basin. We are very much aware that this would be an impossible work. What we want to show is that, using a generalised model, it is possible to determine recharge rates in areas with low accessibility and lack of data.

Consequence: We will clarify these points in the revised manuscript

2. The introduction should be clarified. In particular, a clear presentation of the objective should arrive early in the introduction as a number of different methods are detailed, but their advantages and limits in regards of the objectives of the present study are not clear.

To better organize the introduction, I would recommend

- to first present recharge estimates and the factor controlling it in semi-arid regions (l.78 to 90). Done

- then focus on the case of the LCB (l.30-48) and highlight what is missing and requires further work (objective of the present paper). Done

Consequence: we agree and will implement the suggestions

- In a second part of the introduction, I recommend to gather all descriptions of the existing methods to evaluate the unknown variables on the LCB (recharge, evaporation and transpiration), with their potential and limits of application in the case of scares-data catchments such as the LCB. In particular, the benefit of using both chloride and water contents should be pointed.

Consequence: we will add a short description and highlight the benefit using both chloride concentration and water content.

3. Extreme precipitation events are very important recharge processes in semi-arid regions, which is not taken into account here. Instead of applying the same precipitation rate all days of a month, how would the result be different if irregular precipitation rates were applied with extreme precipitation events?

We did not investigate this point, due to lack of data. However, it has been repeatedly pointed out in the manuscript, e.g.
Lines 425-426: "It is expected that high soil moisture dynamics, rather homogeneous soils, and the monthly resolution of climate data result in a minor impact of soil structure on MVG parametrization and groundwater recharge". Furthermore, in lines 429-430 we write "However, because time resolution of precipitation and evapotranspiration data is monthly, the models probably underestimate soil moisture dynamics"
Lines 438-439: "Extreme rain events that cause surface runoff cannot be reflected in the model".

4. What is the depth of the water table at each soil location? Information such as the thickness of the unsaturated zone at each site are missing. It seems to me that the study is restricted to the first meters of the unsaturated zone, while in this area it can reach up to 30m. I am wondering if the depth of the unsaturated zone investigated here is sufficient to get representative estimates of recharge in the unsaturated zone. I guess the underlying assumption is that there is no ET below the a few meters. If I am correct, the assumption should be clearly stated and discussed. Furthermore, even if water contents and chloride concentrations data are not available deeper, simulations could be run at greater depth.

Depth to groundwater is reported in Table 1. Unsaturated zone varies from 4 m in WL1 to 21 m in ST1 and ST3.
Transpiration depth is limited by the root depth, which reaches a maximum depth of 2.5 m in ST1, the whole profile in ST2, 0.4 m in ST3, 0.5 m in WL1, 0.3 in WL2, and 0.6 m in WL3.
Evaporation enriches the chloride concentration in soil. Therefore, evaporation depth can be estimated observing the vertical profiles of chloride concentration. It corresponds to the depth from which the chloride concentration remains constant. Measured chloride profiles are listed in Tables 2 and 3 of supplement material and graphically shown in Figure 5. Except for ST2, where the chloride profiles seems not to have reached a steady state at 2 m depth, all other profiles show variations only in the first 1-2 m.

Consequence: we will explain our assumption concerning recharge below the root zone more strongly in chapter 3.

5. Please give possible explanations for the discrepancies between simulated and modeled chloride dynamics for ST1 and ST2.

Mean residence time of chloride at both locations are long (109 years) compared to the data availability (49 years for precipitation and 6 years for chloride concentrations. At ST2 the measured profile can only be plausibly modelled with an additional input via ponding water (see chapter 4.3), which gives additional uncertainties.

Consequence: we will add these explanations in the discussion

6. Results on chloride accumulation and retention in soils are very interesting and additional calculations would be interesting. For each profile, what is the mass and mean residence time of chloride stored in soils? What is the concentration of chloride at the bottom of the unsaturated zone? How does it correlate to concentrations measured in groundwater?

The stored chloride mass depends strongly on locations and is time dependent. However, it can be estimated from data shown in Fig 8.
Residence time depends on the soil type, thus max. residence times for the profiles can be estimated from the principle used for setting initial values in the model (Lines 264-268). These results in 106 years for ST1 and ST2, 6 years for ST3, 26 years for WL1 and WL2, and finally 46 years for WL3.
Chloride values measured at the bottom of the soil profiles are comparable to those from groundwater. More precisely:
- ST1: Cl concentration of 0.09 mg/l at 5 m depth in unsaturated zone in 2019. A concentration of 0.338 mg/l was measured in groundwater in December 2016, RWL = 11 m
- ST2: Cl concentration of 0.97 mg/l at 5 m depth in unsaturated zone in 2019. A concentration of 1.39 mg/l was measured in groundwater in December 2016, RWL = 17 m
- ST3: Cl concentration of 0.42 mg/l at 5 m depth in unsaturated zone in 2019. A concentration of 4.1 mg/l was measured in groundwater in December 2016, RWL = 21 m
- WL1: Cl concentration of 0.52 mg/l at 4 m depth in unsaturated zone in 2017. A concentration of 0.225 mg/l was measured in groundwater in the same year, RWL = 4 m
- WL2: Cl concentration of 0.25 mg/l at 3 m depth in unsaturated zone in 2017. A concentration of 0.899 mg/l was measured in groundwater in 2014, RWL = 12 m
- WL3: Cl concentration of 1.9 mg/l at 3.8 m depth in unsaturated zone in 2017. A concentration of 1.51 mg/l was measured in groundwater in the same year, RWL = 3.6 m

Consequence: we will add this information.

7. Is there groundwater data (both chloride concentrations and water levels) that could be used to validate the recharge and chloride fluxes estimates?

Yes, see response to question 6

8. The value of chloride wet deposition of 1.8 ± 0.5 kg ha-1 could also be compared with the value of 1 ± 0.5 kg ha-1. estimated in Bouchez et al., 2019. And the recharge estimated in the present study could also be compared to the recharge estimates in Bouchez et al., 2019 at different locations in the catchment (16 to 240 mm/year).

Chloride wet deposition is compared to Bouchez et al. (2019) in Line 181.
Recharges estimates are compared to Bouchez et al. (2019) in Lines 450-451.

---

## Author Response (AR2)

**Response to the Editor**

You addressed most of the comments adequately and improved the manuscript. However, there are two minor issues remaining, which I think you could improve a bit more. The first being the explanation of the discrepancies between observed and simulated chloride and water contents for ST1 and ST2.

The discrepancies at ST1 and ST2 are not exceptional high (see Table4). However, the dynamics of measured and simulated water contents differ considerably for ST1 and partly for ST2. This is mainly caused by the comparably high evapotranspiration at those locations and the uncertainty of chloride concentration of ponding water.

You did now include values of chloride concentration in the groundwater in S2 and S3, but I couldn't find any way in which you used these values in the article to reflect on your results or to validate the recharge assumptions. So please have a new look at the questions 6 and 7 posed by the second reviewer and consider whether/ how you might use these to further improve your analysis and discussion in the manuscript.

We added information on chloride concentration in groundwater in tables S2 and S3 and in the text (lines 187-188), which we have corrected because they were erroneous. However, these figures are meant for comparison, as the manuscript only models chloride concentrations in the unsaturated zone und does not consider the saturated zone.

Groundwater values are much lower than in soil profiles. This is because the lateral groundwater flow has been recharged in areas, where chloride does not play an important role. Large amount of recharge takes place mainly in the southern part of the Lake Chad Basin, where long-term annual precipitation reaches values over 1500 mm. Any chloride accumulated in the soil is well diluted and washed away periodically.

However, these large differences in chloride concentrations between soils and groundwater in the study area demonstrate the enormous accumulation capacity of soils, which act as a buffer over years until precipitation is large enough to "clean up" the profile, which is well shown in profile ST2 (Figure 7).

This discussion has been added to the manuscript (lines 374-382) and to the response to the second reviewer (questions 6 and 7).

As a minor revision I saw that in line 365 there is a minus sign missing in the superscript of cm3.

Corrected

**Response to Reviewer 2 (Completed, as Required by the Editor**

1.      In the abstract, the authors say that it is a generalized approach. However in the example given here, it actually seems very localized and site-specific. The results are different between each soil, which suggest that we would need a large number of soil profiles to estimate recharge over the catchment. Are the results obtained generalizable? Are the soils and vegetation types studied here covering all expected soils and vegetation types of the LCB? How do the authors extrapolate the local recharge estimation to an average recharge rate?

The sentence reads "A simple, generalized approach, which requires only limited data…". We describe a generalized approach; we do not say that the results are generalizable.

The types of soils we have worked with (sand, loam, clay and their combinations) are the most common in the LCB. However, due to the extension of the LCB, we surely do not cover all existent soils (Lines 157-158). Concerning vegetation, acacia and grass are the most widespread natural vegetation throughout the LCB, whereas sorghum is the most commonly planted corn. Cotton, which is also planted, is only locally produced and generally using irrigation. Mango trees can be found along the Chari and Logone rivers, but are not representative for the whole LCB (Lines 163-166).

We do not intend to extrapolate our values to the whole basin. We are very much aware that this would be an impossible work. What we want to show is that, using a generalised model, it is possible to determine recharge rates in areas with low accessibility and lack of data (Lines 135-136).

2. The introduction should be clarified. In particular, a clear presentation of the objective should arrive early in the introduction as a number of different methods are detailed, but their advantages and limits in regards of the objectives of the present study are not clear.

To better organize the introduction, I would recommend to first present recharge estimates and the factor controlling it in semi-arid regions (l.78 to 90) then focus on the case of the LCB (l.30-48) and highlight what is missing and requires further work (objective of the present paper). In a second part of the introduction, I recommend to gather all descriptions of the existing methods to evaluate the unknown variables on the LCB (recharge, evaporation and transpiration), with their potential and limits of application in the case of scares-data catchments such as the LCB. In particular, the benefit of using both chloride and water contents should be pointed.

We changed the introduction as proposed by the reviewer. Furthermore, we added a short description of applied methods and obtained values for recharge, evaporation, and transpiration in the LCB (Lines 51-96).

3. Extreme precipitation events are very important recharge processes in semi-arid regions, which is not taken into account here. Instead of applying the same precipitation rate all days of a month, how would the result be different if irregular precipitation rates were applied with extreme precipitation events?

We did not investigate this point, due to lack of data. However, it has been repeatedly pointed out in the manuscript, e.g.
Lines 447-449: "It is expected that high soil moisture dynamics, rather homogeneous soils, and the monthly resolution of climate data result in a minor impact of soil structure on MVG parametrization and groundwater recharge". Furthermore, in lines 451-452 we write "However, because time resolution of precipitation and evapotranspiration data is monthly, the models probably underestimate soil moisture dynamics"
Lines 460: "Extreme rain events that cause surface runoff cannot be reflected in the model".

4. What is the depth of the water table at each soil location? Information such as the thickness of the unsaturated zone at each site are missing. It seems to me that the study is restricted to the first meters of the unsaturated zone, while in this area it can reach up to 30m. I am wondering if the depth of the unsaturated zone investigated here is sufficient to get representative estimates of recharge in the unsaturated zone. I guess the underlying assumption is that there is no ET below the a few meters. If I am correct, the assumption should be clearly stated and discussed. Furthermore, even if water contents and chloride concentrations data are not available deeper, simulations could be run at greater depth.

Depth to groundwater is reported in Table 1. Unsaturated zone varies from 4 m in WL1 to 21 m in ST1 and ST3.
Transpiration depth is limited by the root depth, which reaches a maximum depth of 2.5 m in ST1, the whole profile in ST2, 0.4 m in ST3, 0.5 m in WL1, 0.3 in WL2, and 0.6 m in WL3.
Evaporation enriches the chloride concentration in soil. Therefore, evaporation depth can be estimated observing the vertical profiles of chloride concentration. It corresponds to the depth from which the chloride concentration remains constant. Measured chloride profiles are listed in Tables 2 and 3 of supplement material and graphically shown in Figure 5. Except for ST2, where the chloride profiles seems not to have reached a steady state at 2 m depth, all other profiles show variations only in the first 1-2 m.

We explain our assumption concerning recharge below the root zone more strongly in chapter 3. Lines 220-221.

5. Please give possible explanations for the discrepancies between simulated and modeled chloride dynamics for ST1 and ST2.

Mean residence time of chloride at both locations are long (109 years) compared to the data availability (49 years for precipitation and 6 years for chloride concentrations). At ST2 the measured profile can only be plausibly modelled with an additional input via ponding water (see chapter 4.3), which gives additional uncertainties.

We added these explanations in chapter 4.3 (lines 386-387).

6. Results on chloride accumulation and retention in soils are very interesting and additional calculations would be interesting. For each profile, what is the mass and mean residence time of chloride stored in soils? What is the concentration of chloride at the bottom of the unsaturated zone? How does it correlate to concentrations measured in groundwater?

The stored chloride mass depends strongly on locations and is time dependent. However, it can be estimated from data shown in Fig 8.
Residence time depends on the soil type, thus max. residence times for the profiles can be estimated from the principle used for setting initial values in the model (Lines 281-283). These results in 106 years for ST1 and ST2, 6 years for ST3, 26 years for WL1 and WL2, and finally 46 years for WL3 (Lines 284-285).
Chloride values measured at the bottom of the soil profiles are very different to those from groundwater. More precisely:
- ST1: Cl concentration of 1.02 mg/l at 5 m depth in unsaturated zone in 2019. A concentration of 0.34 mg/l was measured in groundwater in December 2016, RWL = 11 m
- ST2: Cl concentration of 149.2 mg/l at 5 m depth in unsaturated zone in 2019. A concentration of 1.39 mg/l was measured in groundwater in December 2016, RWL = 17 m
- ST3: Cl concentration of 39.51 mg/l at 5 m depth in unsaturated zone in 2019. A concentration of 4.10 mg/l was measured in groundwater in December 2016, RWL = 21 m
- WL1: Cl concentration of 6.86 mg/l at 4 m depth in unsaturated zone in 2017. A concentration of 0.23 mg/l was measured in groundwater in the same year, RWL = 4 m
- WL2: Cl concentration of 4.87 mg/l at 3 m depth in unsaturated zone in 2017. A concentration of 0.90 mg/l was measured in groundwater in 2014, RWL = 12 m

- WL3: Cl concentration of 21.78 mg/l at 3.8 m depth in unsaturated zone in 2017. A concentration of 1.51 mg/l was measured in groundwater in the same year, RWL = 3.6 m

We added these informations summarized in tables S2 and S3 and in the text (lines 187-188). These figures are meant for comparison, as the manuscript only models chloride concentrations in the unsaturated zone und does not consider processes in the saturated zone.

7. Is there groundwater data (both chloride concentrations and water levels) that could be used to validate the recharge and chloride fluxes estimates?

The manuscript models chloride concentration in the unsaturated zone und does not consider processes in the saturated zone. Groundwater values are much lower than in soil profiles. This is because the groundwater encountered in the study area has been recharged in zones, where chloride does not play an important role. Large amounts of recharge for the Quaternary aquifer occur mainly in the southern part of the Lake Chad Basin, where long-term annual precipitation reaches values over 1000 mm. Any chloride accumulated in the soil is well diluted and washed away periodically.
However, the large differences in chloride concentrations between soils and groundwater in the study area demonstrate the enormous accumulation capacity of soils, which act as a buffer over years until precipitation is large enough to "clean up" the profile. It is well depicted by the different concentrations in profile ST1 between 2016 and 2019 as well as by the model results in profile ST2 (Figure 7) (lines 374-382)

8. The value of chloride wet deposition of 1.8 ± 0.5 kg ha-1 could also be compared with the value of 1 ± 0.5 kg ha-1 estimated in Bouchez et al., 2019. And the recharge estimated in the present study could also be compared to the recharge estimates in Bouchez et al., 2019 at different locations in the catchment (16 to 240 mm/year).

Chloride wet deposition is compared to Bouchez et al. (2019) in Line 194.
Recharges estimates are compared to Bouchez et al. (2019) in Lines 456-457.